# Notch-induced endoplasmic reticulum-associated degradation governs mouse thymocyte β−selection

Xia Liu[1,2†], Jingjing Yu[1,2†], Longyong Xu[1,2†], Katharine Umphred-Wilson[3†], Fanglue Peng[1,2], Yao Ding[1,2], Brendan M Barton[3], Xiangdong Lv[1], Michael Y Zhao[1], Shengyi Sun[4], Yuning Hong[5], Ling Qi[6], Stanley Adoro[3]*, Xi Chen[1,2]*

[1]Department of Molecular and Cellular Biology, Baylor College of Medicine, Houston, United States; [2]Lester and Sue Smith Breast Center and Dan L Duncan Comprehensive Cancer Center, Baylor College of Medicine, Houston, United States; [3]Department of Pathology, School of Medicine, Case Western Reserve University, Cleveland, United States; [4]Center for Molecular Medicine and Genetics, Wayne State University, Detroit, United States; [5]Department of Chemistry and Physics, La Trobe University, Melbourne, Australia; [6]Department of Molecular and Integrative Physiology, University of Michigan Medical School, Ann Arbor, United States

*For correspondence:
sxa726@case.edu (SA);
xi.chen@bcm.edu (XC)

†These authors contributed equally to this work

Competing interests: The authors declare that no competing interests exist.

**Abstract** Signals from the pre-T cell receptor and Notch coordinately instruct β-selection of CD4⁻CD8⁻double negative (DN) thymocytes to generate αβ T cells in the thymus. However, how these signals ensure a high-fidelity proteome and safeguard the clonal diversification of the pre-selection TCR repertoire given the considerable translational activity imposed by β-selection is largely unknown. Here, we identify the endoplasmic reticulum (ER)-associated degradation (ERAD) machinery as a critical proteostasis checkpoint during β-selection. Expression of the SEL1L-HRD1 complex, the most conserved branch of ERAD, is directly regulated by the transcriptional activity of the Notch intracellular domain. Deletion of *Sel1l* impaired DN3 to DN4 thymocyte transition and severely impaired mouse αβ T cell development. Mechanistically, *Sel1l* deficiency induced unresolved ER stress that triggered thymocyte apoptosis through the PERK pathway. Accordingly, genetically inactivating PERK rescued T cell development from *Sel1l*-deficient thymocytes. In contrast, IRE1α/XBP1 pathway was induced as a compensatory adaptation to alleviate *Sel1l*-deficiency-induced ER stress. Dual loss of *Sel1l* and *Xbp1* markedly exacerbated the thymic defect. Our study reveals a critical developmental signal controlled proteostasis mechanism that enforces T cell development to ensure a healthy adaptive immunity.

## Introduction

T cells develop from bone-marrow-derived early T-cell progenitors (ETP) through a series of well-orchestrated proliferation and differentiation steps in the thymus. In response to intrathymic interleukin (IL)−7 and Kit ligand, ETPs proliferate and differentiate into CD4⁻CD8⁻double negative (DN) thymocytes (*von Freeden-Jeffry et al., 1997*). Subsequent differentiation of DN thymocytes into CD4⁺CD8⁺ (double positive, DP) thymocytes depends on whether DN3 stage (CD44⁻CD25⁺) thymocytes successfully undergo 'β-selection', the first major checkpoint during αβ T cell development (*Shah and Zúñiga-Pflücker, 2014*). β-selection is initiated by signals from the pre-TCR (a heterodimer of the invariant pre-Tα and TCRβ proteins) in DN3 thymocytes that have productively undergone V(D)J recombination at the *Tcrb* locus (*Mallick et al., 1993*; *Michie and Zúñiga-Pflücker,*

*2002*). In addition to cell autonomous signal through the pre-TCR, β-selection also requires signal from the Notch receptor (*Ciofani and Zúñiga-Pflücker, 2005*; *Sambandam et al., 2005*). Coordinately, pre-TCR and Notch signals induce DN3 thymocytes to undergo 100–200 fold clonal expansion (*Yamasaki et al., 2006*; *Zhao et al., 2019*) as they differentiate into DN4 (CD44⁻CD25⁻) cells which give rise to the DP thymocyte precursors of mature αβ T cells. This proliferative burst is crucial for the diversification of the pre-selection TCR repertoire (*Kreslavsky et al., 2012*) and so must be robustly buffered to ensure adequate number of thymocytes audition for positive selection. β-selection imposes a considerable demand for new protein synthesis of the newly rearranged *Tcrb* gene and the multiple factors that execute the transcriptional and metabolic programs demanded by DN thymocyte proliferation. However, how proteome homeostasis or 'proteostasis' is regulated during thymocyte development is largely unknown.

Endoplasmic reticulum (ER) is the major subcellular site for synthesis and maturation of all transmembrane and secreted proteins. Protein folding is an inherently error-prone process and is tightly regulated by a myriad of chaperones and enzymes (*Balchin et al., 2016*; *Cox et al., 2020*). To maintain proteostasis and normal cell function, cells have evolved highly sensitive and sophisticated quality control systems to ensure the fidelity of protein structure, which is especially important for thymocytes undergoing β-selection that must repair protein damage and generate a functional and diverse repertoire of T cell receptors with high fidelity (*Feige et al., 2015*; *Feige and Hendershot, 2013*). Two such systems conserved across different species are ER-associated degradation (ERAD) and the unfolded protein response (UPR) (*Figure 1—figure supplement 1A*; *Brodsky, 2012*; *Hwang and Qi, 2018*; *Walter and Ron, 2011*). ERAD is the principal protein quality control mechanism responsible for targeting misfolded proteins in the ER for cytosolic proteasomal degradation. The E3 ubiquitin ligase HRD1 and its adaptor protein SEL1L, constitute the most conserved branch of ERAD (*Brodsky, 2012*; *Qi et al., 2017*; *Ruggiano et al., 2014*; *Sun et al., 2014*). SEL1L recruits misfolded proteins bound by ER protein chaperones to the SEL1L-HRD1 complex, through which the misfolded proteins are retrotranslocated into cytosol, ubiquitinated and degraded by the proteasome in the cytosol with the help of CDC48/ p97 (*Brodsky, 2012*; *Nakatsukasa and Brodsky, 2008*). Failure to clear the misfolded proteins in the ER activates the UPR (*Brodsky, 2012*; *Hwang and Qi, 2018*; *Ruggiano et al., 2014*). The UPR is a highly conserved, three-pronged pathway that is activated when the rate of cellular protein production exceeds the capacity of the ER to correctly fold and process its protein load or by various intracellular and extracellular stressors that interfere with the protein folding process. This coordinated response is mediated by three ER-localized transmembrane sensors: IRE1α, ATF6α, and PERK (*Hetz et al., 2011*). Under ER stress, IRE1α undergoes oligomerization and *trans*-autophosphorylation to activate its RNase domain to induce unconventional splicing of its substrate XBP1 (*Hwang and Qi, 2018*; *Walter and Ron, 2011*). ER stress also induces PERK-dependent eIF2α phosphorylation and subsequent increased cap-independent translation of ATF4 and induction of CHOP (*Figure 1—figure supplement 1A*; *Hwang and Qi, 2018*; *Walter and Ron, 2011*).

Here, we show that ERAD is the master regulator of physiological ER proteostasis in immature DN thymocytes. The ERAD machinery was critically required for successful β-selection of DN3 thymocytes and consequently, ERAD deficiency impeded αβ T cell development. Intriguingly, ERAD selectively preserves the cellular fitness of αβ, but not γδ T lymphocytes. We found that Notch signaling directly regulates ERAD gene expression to promote the integrity of ER proteostasis during β-selection. Activation of ERAD restricts PERK-dependent cell death in DN3 thymocytes during β-selection. Genetic inactivation of *Perk* rescued β-selection in *Sel1l*-deficient thymocytes.

## Results

### Stringent protein quality control in β-selected thymocytes

To determine translational dynamics in developing thymocytes, we injected wildtype C57BL/6 (WT) mice with O-propargyl puromycin (OP-Puro), a cell-permeable puromycin analog that is incorporated into newly synthesized proteins as a measure of protein synthesis rates (*Hidalgo San Jose and Signer, 2019*; *Tong et al., 2020*). Animals were euthanized 1 hr after OP-Puro injection, and thymocyte subsets (gated as shown in *Figure 1—figure supplement 1B,C*) were assessed by flow cytometry for OP-Puro incorporation (*Figure 1A*). Compared to DP and mature single positive thymocytes,

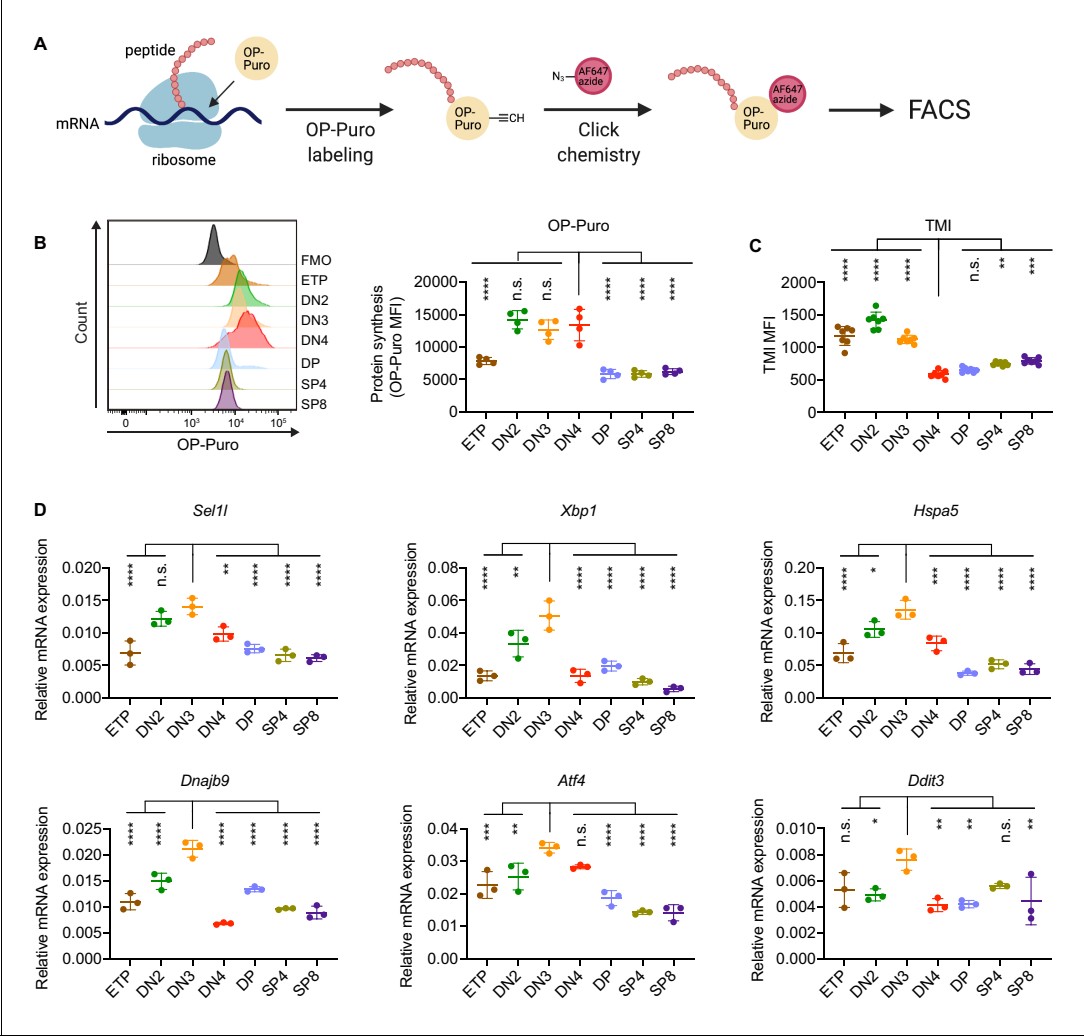

**Figure 1.** Protein quality control in β-selected thymocytes. (**A**) Schematic of labeling and detection of nascent protein with OP-Puro. OP-Puro (O-propargyl puromycin) is a cell-permeable puromycin analog that is incorporated into the C-terminus of newly synthesized peptide chain. Fluorophore conjugated with Alexa Fluor 647 was then attached to OP-Puro through a copper-catalyzed click chemistry reaction between alkyne and azide group, which quantifies protein synthesis by fluorescence intensity. (**B**) Representative histogram (left) and quantification (right) of OP-Puro incorporation in different thymocyte subsets from 8-week-old wild-type mice. FMO represents AF647 control which is the background from the click chemistry in the absence of OP-Puro. MFI, mean fluorescence intensity. $n$ = four mice. (**C**) Quantification of tetraphenylethene maleimide (TMI) fluorescence in different thymocyte subsets from 8-week-old wild-type mice. $n$ = seven mice. (**D**) Quantitative RT-PCR analysis of ERAD (*Sel1l*) and UPR-related (*Xbp1*, *Hspa5 (Bip)*, *Dnajb9*, *Ddit3* (*Chop*), *Atf4*) genes expression in different thymocyte subsets from 6-week-old wild-type mice. Data are presented relative to *Actb*; $n$ = three mice. (**B–D**), ETP: early T lineage precursor (Lin$^-$ CD4$^-$ CD8$^-$ CD44$^+$ CD25$^-$ CD117$^+$); DN2: double negative two thymocytes (Lin$^-$ CD4$^-$ CD8$^-$ CD44$^+$ CD25$^+$); DN3: double negative three thymocytes (Lin$^-$ CD4$^-$ CD8$^-$ CD44$^-$ CD25$^+$); DN4: double negative four thymocytes (Lin$^-$ CD4$^-$ CD8$^-$ CD44$^-$ CD25$^-$); DP: double positive thymocytes (Lin$^-$ CD4$^+$ CD8$^+$); SP4: CD4 single positive thymocytes (Lin$^-$ CD4$^+$ CD8$^-$); SP8: CD8 single positive thymocytes (Lin$^-$ CD4$^-$ CD8$^+$). Results are shown as mean ± s.d. The statistical significance was calculated by one-way ANOVA with Bonferroni test. *p < 0.05, **p < 0.01, ***p < 0.001, ****p < 0.0001, n.s., not significant.

The online version of this article includes the following source data and figure supplement(s) for figure 1:

**Source data 1.** Excel file containing numerical values shown in *Figure 1*.

**Figure supplement 1.** Diagrams and representative flow cytometry gates used in this study.

DN2 to DN4 thymocytes incorporated the most OP-Puro (*Figure 1B*), a likely reflection of their high metabolic and proliferative activity (*Carpenter and Bosselut, 2010*; *Kreslavsky et al., 2012*; *Nagelreiter et al., 2018*). To assess the relationship between translational activity and proteome quality, we stained WT thymocytes with tetraphenylethene maleimide (TMI), a cell-permeable reagent that only fluoresces when bound to free thiol groups typically exposed on misfolded or

unfolded proteins (*Chen et al., 2017*; *Hidalgo San Jose et al., 2020*). Intriguingly, despite comparable and high protein synthesis rates across DN2 to DN4 thymocytes, we found that DN4 thymocytes displayed markedly lower levels of misfolded/unfolded proteins (*Figure 1C*). These observations suggested that the DN3-to-DN4 transition, which is initiated by β-selection, is accompanied by induction of proteome quality control mechanisms.

To understand the protein quality control mechanisms operating in thymocytes, we performed quantitative PCR to determine expression of genes encoding ER protein quality control machinery. We found elevated expression of the core ERAD (*Sel1l*) and the UPR (*Xbp1*, *Ddit3 (Chop)*, *Atf4*, *Dnajb9*, and *Hspa5 (Bip)*) genes in DN3 thymocytes (*Figure 1D*). Notably, induction of these genes peaked in the DN3 thymocyte stage in which β-selection is initiated (*Takahama, 2006*) and preceded the reduction of misfolded/unfolded proteins in DN4 cells. These results prompted us to hypothesize and explore whether β-selection signals induce proteome quality control mechanisms in DN3 cells to enable subsequent stages of thymocyte development.

## The ERAD machinery is required for αβ T cell development

To resolve the ER proteostasis machinery required for the development of thymocytes, we conditionally deleted individual genes encoding key mediators of the UPR (*Xbp1*, *Perk*) and ERAD (*Sel1l*) using the h*CD2*-iCre transgene (*Siegemund et al., 2015*). In this model, significant Cre activity initiated in ETP thymocytes (*Shi and Petrie, 2012*; *Siegemund et al., 2015*) and was efficient in depleting genes in subsequent stages of thymocyte development and mature T cells (*Figure 2—figure supplement 1A,B*). Deletion of the UPR mediator *Xbp1* or *Perk* had no effect on T cell development as thymic cellularity, numbers of DN, DP, single positive (SP) thymocytes and splenic T cell numbers were comparable in gene-deficient and littermate control animals (*Figure 2—figure supplement 1C–J*). Similarly, *Vav1*-iCre-mediated deletion of *Atf6a* which initiates in bone marrow hematopoietic cell progenitors (*Joseph et al., 2013*) did not perturb thymocyte development (*Figure 2—figure supplement 1K–N*).

Strikingly, *CD2*-iCre-mediated deletion of the ERAD core component *Sel1l* (*Sel1l*<sup>flox/flox</sup>; CD2-iCre mice, hereafter designated as *Sel1l* CKO) resulted in a markedly decreased thymus size and cellularity, with significantly reduced and dispersed medullary regions compared to control littermates (*Sel1l*<sup>flox/flox</sup>; Ctrl) (*Figure 2A–D*). The reduced thymus cellularity was accompanied by a profound reduction of peripheral T cells in the spleen and lymph nodes from *Sel1l* CKO mice compared to control animal (*Figure 2—figure supplement 1O–R*). *Sel1l* deletion had no impact on γδ T cells (*Figure 2—figure supplement 1S*), indicating that within the T-cell lineage, SEL1L is selectively required for αβ T cell development.

## SEL1L is required for DN to DP thymocyte transition following β selection

While *Sel1l* CKO mice had similar numbers of DN thymocytes (*Figure 2E* and *Figure 2—figure supplement 2A*), they showed significantly reduced numbers of DP and mature SP thymocytes compared to littermate controls (*Figure 2F*). This finding suggests that SEL1L functioned during the DN to DP thymocyte transition. To clarify this possibility, we generated and analyzed thymocyte developmental stages in *Sel1l*<sup>flox/flox</sup>; *Cd4*-Cre mice. Unlike *CD2*-iCre which initiated in ETP (*Siegemund et al., 2015*), *Cd4*-Cre initiated in immature single-positive (ISP, CD8<sup>+</sup>CD24<sup>+</sup>TCRβ<sup>-</sup>) thymocytes (*Gegonne et al., 2018*; *Kadakia et al., 2019*; *Xu et al., 2016*) and only significantly depleted *Sel1l* in ISPs and later stage thymocytes (*Figure 2—figure supplement 2B,C*). *Sel1l*<sup>flox/flox</sup>; *Cd4*-Cre mice exhibited indistinguishable thymic cellularity, immature DN, DP and SP thymocytes numbers from control mice (*Figure 2—figure supplement 2D–G*). Thus, whereas SEL1L is dispensable for differentiation of post-DN4 thymocytes (i.e. DP, SP, and mature T cells), its expression is critical for DN to DP thymocyte differentiation.

To delineate the DN thymocyte developmental stage at which SEL1L is required, we generated 1:1 mixed bone marrow (BM) chimeras by transplanting equal numbers of whole BM cells from control (*Sel1l*<sup>flox/flox</sup>, CD45.2<sup>+</sup>) or *Sel1l* CKO (*Sel1l*<sup>flox/flox</sup>; *CD2*-iCre, CD45.2<sup>+</sup>) mice along with congenic (CD45.1<sup>+</sup>) wild-type (WT) competitor BM cells into irradiated CD45.1<sup>+</sup> recipient mice (*Figure 2G*). Fourteen weeks after transplantation, control and *Sel1l* CKO donors reconstituted similar numbers of all BM hematopoietic progenitors including Lineage<sup>-</sup>Sca-1<sup>+</sup>c-Kit<sup>+</sup> (LSK) cells, hematopoietic stem

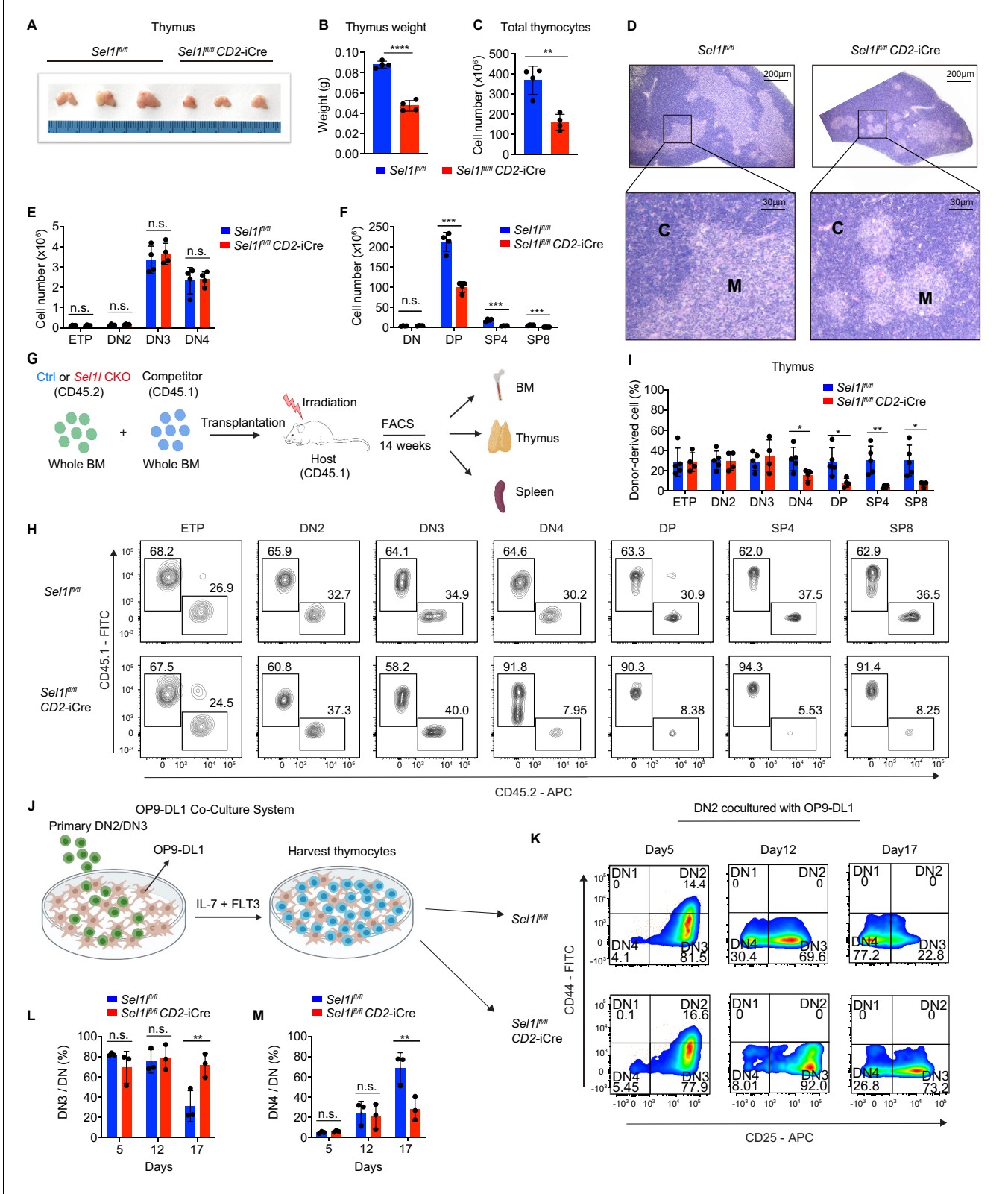

**Figure 2.** SEL1L is required for αβ T cell development. (**A**) Images of thymus from 6 to 8 week-old control (Ctrl, *Sel1l*<sup>flox/flox</sup>) and *Sel1l* CKO (*Sel1l*<sup>flox/flox</sup>; *CD2*-iCre) mice. *n* = 3. (**B and C**) Thymus weight (**B**) and thymus cellularity (**C**) of age and gender-matched control (Ctrl, *Sel1l*<sup>flox/flox</sup>) and *Sel1l* CKO (*Sel1l*<sup>flox/flox</sup>; *CD2*-iCre) mice. *n* = 4. (**D**) Representative images of H and E staining of thymus from 6~8-week-old control (Ctrl, *Sel1l*<sup>flox/flox</sup>) and *Sel1l* CKO (*Sel1l*<sup>flox/flox</sup>; *CD2*-iCre) mice. Scale bars are indicated. C: Cortex. M: Medulla. (**E and F**) Quantification of cell numbers of the indicated thymocyte

*Figure 2 continued on next page*

eLife Research article

Developmental Biology | Immunology and Inflammation

*Figure 2 continued*

subsets in 6- to 8-week-old control (Ctrl, *Sel1l*^*flox/flox*) and *Sel1l* CKO (*Sel1l*^*flox/flox*; *CD2*-iCre) mice. *n* = 4. (**G**) Schematic depiction of the competitive bone marrow transplantation (BMT) experiment using whole bone marrow cells from control (Ctrl, *Sel1l*^*flox/flox*) or *Sel1l* CKO (*Sel1l*^*flox/flox*; *CD2*-iCre) mice as donors. (**H and I**) Representative flow cytometry plots (**H**) and percentage (**I**) of control (Ctrl, *Sel1l*^*flox/flox*) or *Sel1l* CKO donor-derived thymocyte subsets in the recipient mice 14 weeks after transplantation. *n* = 4–5. (**J**), Schematic overview of OP9-DL1 cell co-culture system. Sorted DN2 or DN3 cells from control (Ctrl) or *Sel1l* CKO mice were cultured on a monolayer of OP9 -DL1 cells supplemented with IL-7 and Flt3. (**K, L, M**) Representative pseudocolor plots (**K**) and percentage of DN3 (**L**) or DN4 (**M**) in DN thymocytes at indicated time points after in vitro co-culture of equal number of control (Ctrl, *Sel1l*^*flox/flox*) or *Sel1l* CKO DN2 cells on OP9-DL1 cells supplemented with IL-7 and Flt3. *n* = 3. Results are shown as mean ± s.d. The statistical significance was calculated by two-tailed unpaired t-test (**B, C, E, F, I**) or two-way ANOVA with Bonferroni test (**L, M**). *$p < 0.05$, **$p < 0.01$, ***$p < 0.001$, ****$p < 0.0001$, n.s., not significant.

The online version of this article includes the following source data and figure supplement(s) for figure 2:

**Source data 1.** Excel file containing numerical values shown in *Figure 2*.

**Figure supplement 1.** UPR is dispensable for αβ T cell development.

**Figure supplement 1—source data 1.** Original western blot images shown in *Figure 2—figure supplement 1*.

**Figure supplement 1—source data 2.** Excel file containing numerical values shown in *Figure 2—figure supplement 1*.

**Figure supplement 2.** SEL1L is required for DN to DP thymocyte transition following β selection.

**Figure supplement 2—source data 1.** Excel file containing numerical values shown in *Figure 2—figure supplement 2*.

progenitor cells (HSPC), multipotent progenitor (MPP), and myeloid progenitors (Lineage⁻Sca-1⁻c-Kit⁺LS-K) cells (*Figure 2—figure supplement 2H*). We found substantial defective thymus

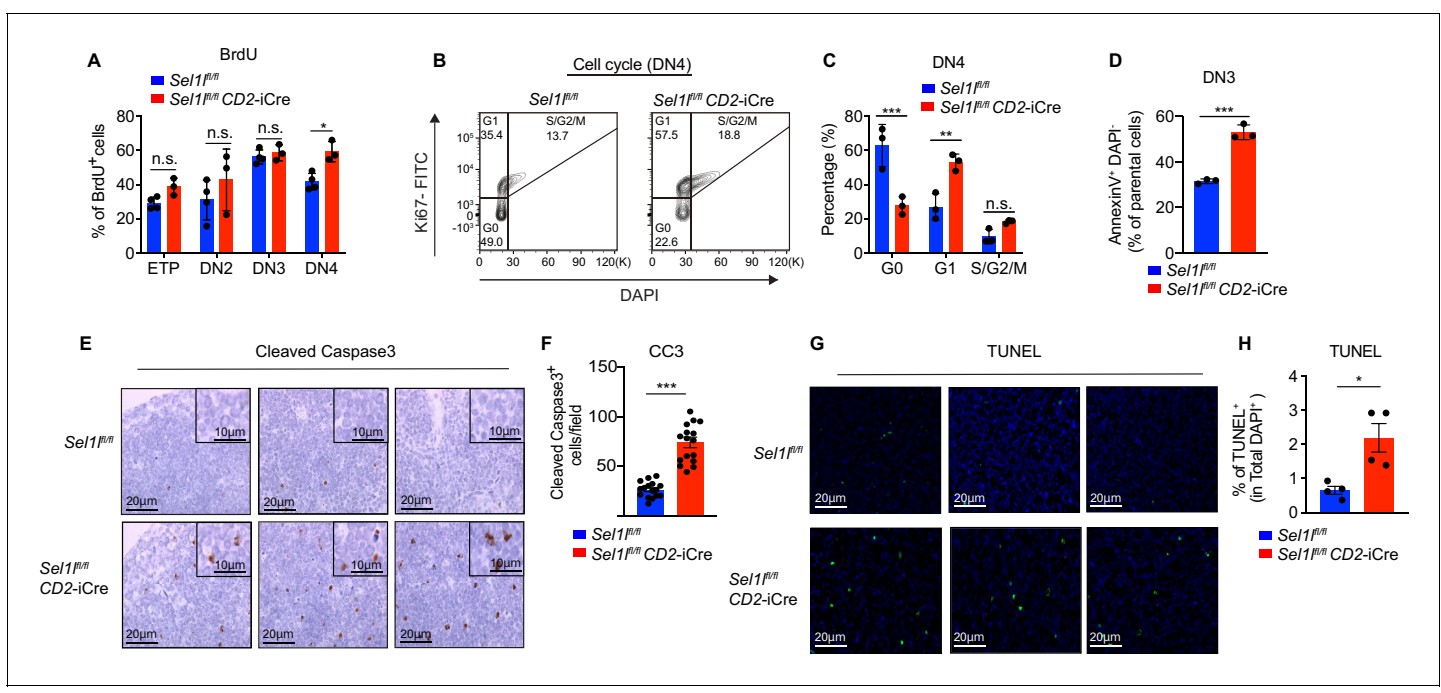

**Figure 3.** SEL1L is required for thymocyte survival at the β-selection checkpoint. (**A**) Quantification of BrdU incorporation in different thymocyte subsets from 6-week-old control (Ctrl, *Sel1l*^*flox/flox*) or *Sel1l* CKO mice. *n* = 3–4. (**B and C**) Cell cycle analysis of DN4 thymocytes in 6-week-old control (Ctrl, *Sel1l*^*flox/flox*) and *Sel1l* CKO (*Sel1l*^*flox/flox*; *CD2*-iCre) using Ki67 and DAPI. Representative flow cytometry plots (**B**) and quantification (**C**) are shown. *n* = 3. (**D**) Quantification of apoptotic Ctrl or *Sel1l* CKO (*Sel1l*^*flox/flox*; *CD2*-iCre) DN3 thymocytes co-cultured with OP9-DL1 cells in vitro for 2 days. *n* = 3. (**E and F**) Representative images (**E**) and quantification (**F**) of cleaved caspase-3 (CC3) positive cells in the thymus of 6- to 8-week-old control (Ctrl, *Sel1l*^*flox/flox*) or *Sel1l* CKO (*Sel1l*^*flox/flox*; *CD2*-iCre) mice. Sixteen fields were counted at ×20 magnification from 4 Ctrl or *Sel1l* CKO mice. Scale bars are indicated. (**G and H**) Representative images (**G**) and quantification (**H**) of TUNEL positive cells in the thymus of 6- to 8-week-old control (Ctrl, *Sel1l*^*flox/flox*) or *Sel1l* CKO (*Sel1l*^*flox/flox*; *CD2*-iCre) mice. *n* = 4. Scale bar, 20 μM. Results are shown as mean ± s.d. The statistical significance was calculated by two-tailed unpaired t-test (**D, F, H**) or two-way ANOVA with Bonferroni test (**A, C**). *$p < 0.05$, **$p < 0.01$, ***$p < 0.001$, ****$p < 0.0001$, n.s., not significant.

The online version of this article includes the following source data for figure 3:

**Source data 1.** Excel file containing numerical values shown in *Figure 3*.

reconstitution starting from the DN4 stage from *Sel1l*-CKO donors (*Figure 2H,I*). *Sel1l* CKO-donor-derived DN3 to DN4 ratio was significantly increased compared to controls (*Figure 2—figure supplement 2I*), suggesting that *Sel1l* depletion compromised the fitness of DN3 thymocytes in which β-selection occurs and impaired their transition to the DN4 stage. The impaired DN thymus reconstitution from *Sel1l* CKO donors was accompanied by severe defects in subsequent donor-derived DP and SP thymocytes (*Figure 2H,I*), as well as peripheral T cells in the spleen (*Figure 2—figure supplement 2J*). These data demonstrate a cell-intrinsic requirement of *Sel1l* for DN3 thymocyte progression to later stages of thymocyte differentiation. *Sel1l* deletion also caused significant B cell reconstitution defects (*Figure 2—figure supplement 2J*), consistent with previous publications showing that SEL1L/HRD1 ERAD is required for B cell development through regulating pre-BCR (*Ji et al., 2016*; *Yang et al., 2018*).

To further clarify the requirement for SEL1L in DN3-to-DN4 thymocyte transition during β-selection, we used the in vitro T cell differentiation system of culturing immature thymocytes on monolayers of OP9 stromal cells expressing the Notch ligand Delta-like 1 (OP9-DL1 cells) supplemented with IL-7 and Flt3 ligand (*Balciunaite et al., 2005*; *Holmes and Zúñiga-Pflücker, 2009*; *Schmitt et al., 2004*; *Figure 2J*). We cultured DN2 thymocytes from control or *Sel1l* CKO mice on OP9-DL1 cells and found that *Sel1l*-deficiency markedly abrogated DN4 thymocyte generation (*Figure 2K–M*). On day 17 of co-culture, only 31.3% DN3 cells remained in control-derived cells, while more than three-fold DN3 cells (71.6%) were found in *Sel1l* CKO-derived cells (*Figure 2K–M*), consistent with a block in DN3 to DN4 thymocyte transition. Since only DN3 thymocytes that have successfully undergone β-selection differentiate to the DN4 and DP thymocyte stage, these results reinforce that cell-intrinsic SEL1L activity is required for post β-selection DN thymocyte development.

## SEL1L is required for thymocyte survival at the β-selection checkpoint

Next, we assessed the impact of *Sel1l* deletion on DN thymocyte proliferation and survival, two key outcomes of successful β-selection (*Ciofani and Zúñiga-Pflücker, 2005*; *Kreslavsky et al., 2012*). Although BrdU incorporation was similar in control and *Sel1l* CKO DN3 thymocytes, we observed more BrdU incorporation in *Sel1l* CKO DN4 thymocytes than that in control DN4 thymocytes (*Figure 3A*). In agreement, co-staining for Ki-67 and the DNA dye (DAPI) revealed less *Sel1l* CKO DN4 thymocytes in G0 phase, and more in G1 and S/G2/M phase (*Figure 3B,C*). *Sel1l* CKO DN3 cells showed similar cell cycle kinetics as control DN3 thymocytes, in line with their normal levels of BrdU incorporation (*Figure 2—figure supplement 2K,L*). The higher proliferation of endogenous *Sel1l*-deficient DN4 thymocytes may be a compensation for their compromised fitness. This likely explains the paradoxical observation that *Sel1l* CKO donors generated markedly reduced DN4 thymocytes in BM chimeras (*Figure 2H,I*), yet *Sel1l* CKO mice had comparable DN4 thymocytes as control mice in steady state (*Figure 2E*).

That, despite their higher proliferative status, *Sel1l* CKO DN4 thymocytes failed to progress to DP thymocytes and generate mature T-cells prompted us to ask if *Sel1l* deficiency resulted in apoptosis of β-selected thymocytes. To test this possibility, we cultured equal number of DN3 thymocytes from control or *Sel1l* CKO mice on OP9-DL1 stromal cells and assessed their apoptosis in vitro. Indeed, *Sel1l* CKO DN3 thymocytes showed higher apoptosis measured by proportions of Annexin-V positive cells (*Figure 3D*). To confirm that *Sel1l*-deficiency resulted in DN thymocyte apoptosis in vivo, we histologically enumerated apoptosis in thymic sections by measuring active Caspase-3 which degrades multiple cellular proteins and is responsible for morphological changes and DNA fragmentation in cells during apoptosis (*Bai et al., 2013*). Compared to controls, *Sel1l* CKO thymus showed more active Caspase-3 apoptotic cells in the cortex area (*Figure 3E,F*), which is typically populated by DN thymocytes (*Trampont et al., 2010*). This observation was further confirmed using TUNEL (terminal deoxynucleotidyl transferase dUTP nick end labeling) staining (*Figure 3G,H*). Taken together, we concluded that *Sel1l*-deficient β-selected DN3 cells were undergoing apoptosis as they differentiated into DN4 thymocytes.

## Notch directly regulates transcription of ERAD genes

Having established that the SEL1L-ERAD is a crucial proteostasis machinery required for β-selection, we next sought to understand the thymic signals which regulate *Sel1l* expression in DN thymocytes.

Because high *Sell1l* expression (*Figure 1D*) coincided with high levels of Notch1 in DN2 and DN3 thymocytes (*Figure 4A*), we explored the possibility that Notch ligands might activate the ERAD machinery to enable DN thymocytes to maintain proteostasis during β-selection. Stimulation of the thymoma cell line EL4 with Notch ligand Delta-like 4 (DLL4) (*Camelo et al., 2012*; *Fitzgerald and Greenwald, 1995*) induced expression of the genes involved in ERAD including *Sel1l*, *Hrd1*, *Os9*, and *Edem1* as well as SEL1L proteins (*Figure 4B,C*). Induction of these genes by DLL4 was concomitant with the induction of classical Notch targets like *Hes1*, *Deltex1*, and *Ptcra* (pre-Tα) (*Figure 4—figure supplement 1A*).

To further determine whether Notch regulates expression of ERAD component genes, we treated freshly isolated primary DN3 thymocytes with the highly specific γ-secretase inhibitor DAPT (N-[N-(3,5-difluorophenacetyl)-L-alanyl]-S-phenylglycine t-butyl ester) that blocks ligand-induced cleavage of the Notch intracellular domain (NICD), preventing its nuclear translocation and subsequent transactivation of the RBP-J transcription factor at target genes (*Chen et al., 2019*; *De Obaldia et al., 2013*; *Schmitt et al., 2004*; *Bray, 2016*). Indeed, treatment with DAPT significantly reduced expression of Notch target genes *Hes1*, *Deltex1*, and *Ptcra* (*Figure 4—figure supplement 1B*) and also reduced *Sel1l*, *Hrd1*, *Os9*, and *Edem1* expression in DN3 thymocytes (*Figure 4D*). These data demonstrate that Notch signals regulate expression of genes constituting the ERAD machinery.

Analysis of the promoters of the core ERAD genes, *Sel1l* and *Hrd1*, revealed conserved binding sites for RBP-J (*Figure 4E*), the DNA binding partner and master transcription factor of the NICD transactivation complex (*Castel et al., 2013*; *Tanigaki and Honjo, 2007*). To interrogate how Notch regulates the ERAD gene components, we performed chromatin immunoprecipitation and qPCR of target DNA (ChIP-qPCR) experiments in EL4 cells. DLL4 stimulation of Notch signaling significantly induced NICD and RBP-J bindings at the promoters of *Sel1l*, *Hrd1*, *Os9*, and *Edem* (*Figure 4F–I*).

We cloned the *Sel1l* and *Hrd1* promoters containing the RBP-J-binding sites into the pGL3 firefly luciferase reporter and tested the regulation of these promoters by Notch. When co-transfected with promoter luciferase reporters into HEK293T cells, NICD potently induced both *Sel1l* and *Hrd1* promoter activity in a dose-dependent manner (*Figure 4J*). In agreement, DLL4 stimulation of Notch signaling in EL4 cells also substantially activated *Sel1l* and *Hrd1* promoter activity (*Figure 4K,L*). Importantly, mutation of the RBP-J-binding sites abolished DLL4-driven induction of *Sel1l* or *Hrd1* luciferase reporter activity (*Figure 4K,L*). These data indicate that Notch signaling directly regulates expression of SEL1L ERAD machinery. Interestingly, although ERAD is known to regulate surface receptor expression in a substrate-specific manner (*van den Boomen and Lehner, 2015*; *Xu et al., 2020*), *Sel1l* deletion did not affect surface Notch1 protein levels in DN thymocyte populations (*Figure 4—figure supplement 1C*). Similarly, SEL1L had little impact on the expression of Notch target genes *Hes1* and *Ptcra* in DN3 thymocytes (*Figure 4—figure supplement 1D*). These results not only suggest that ERAD does not regulate Notch signaling per se but also imply that thymocyte β-selection defects in *Sel1l CKO* mice were not due to failure in Notch signal transduction.

## SEL1L is not required for pre-TCR signaling

The *Tcrb* allele is rearranged in DN2/DN3 thymocytes and the resulting TCRβ protein pairs with pre-Tα and CD3 complex proteins to form the pre-TCR which, together with Notch, transduces β-selection signals and promotes survival, proliferation and further differentiation of DN3 thymocytes (*Ciofani et al., 2004*; *Michie and Zúñiga-Pflücker, 2002*; *Sambandam et al., 2005*). DN3 thymocytes that fail to undergo productive recombination of the *Tcrb* locus fail β-selection, do not complete the DN4-DP transition and are eliminated by apoptosis (*Ciofani et al., 2004*). Therefore, to understand how SEL1L regulates β-selection thymocyte survival and the resulting DN-to-DP transition, we first asked whether *Sel1l*-deficiency caused defective V(D)J recombination. Genomic DNA analysis of *Sel1l*-deficient DN3 and DN4 thymocytes showed that recombination of *Vb5-Jb2*, *Vb8-Jb2*, and *Vb11-Jb2* gene segments were not altered (*Figure 5—figure supplement 1A*), indicating that *Sel1l* deficiency does not affect *Tcrb* gene rearrangement.

Next, we asked whether SEL1L regulates the expression and signaling of the pre-TCR complex. Expression of intracellular TCRβ in pre-selected DN3a, post-selected DN3b and DN4 cells was comparable between control and *Sel1l* CKO mice (*Figure 5—figure supplement 1B,C*). *Sel1l* deletion also had no impact on the expression of pre-TCR signaling intermediates including LCK and ZAP-70 in DN3 and DN4 thymocytes (*Figure 5—figure supplement 1D*).

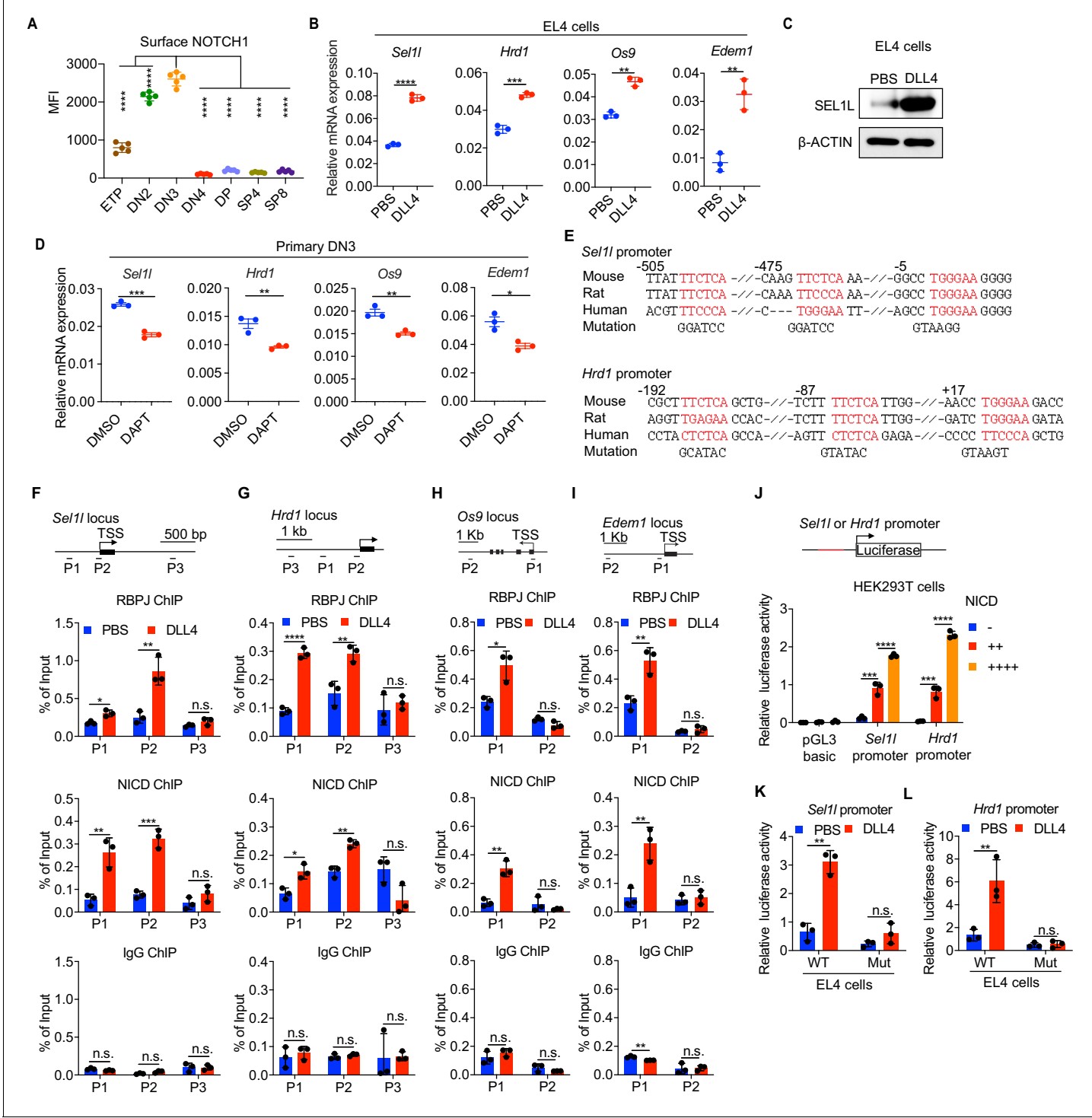

**Figure 4.** Notch directly regulates transcription of ERAD genes. (**A**) Quantification of surface NOTCH1 levels in different thymocyte subsets from wild-type mice. MFI, mean fluorescence intensity. *n* = four mice. (**B**) Quantitative RT–PCR analysis of ERAD genes (*Sel1l*, *Hrd1*, *Os9*, *Edem1*) expression in EL4 cells after stimulation with 5 μg/ml Delta ligand 4 (DLL4) for 24 hr. Data are presented relative to *Actb*. *n* = 3. (**C**) Western blot analysis of SEL1L level in EL4 cells after stimulation with Delta ligand 4 (DLL4) for 12 hr. β-ACTIN was used as loading control. The original western blot images are provided in *Figure 4—source data 1*. (**D**) Quantitative RT–PCR analysis of ERAD genes (*Sel1l*, *Hrd1*, *Os9*, *Edem1*) expression in primary DN3 thymocytes treated with 2 μM γ-secretase inhibitor DAPT for 5 hr. Data are presented relative to *Actb*. *n* = 3. (**E**) Conserved RBP-J binding motif (Red) within the promoters of *Sel1l* and *Hrd1*. Alignment of the *Sel1l* (Upper) or *Hrd1* (lower) promoter from genomic sequence from human, mouse, and rat. The numbering corresponds to the mouse sequence and is relative to the transcription start site (TSS). Mutations of the RBP-J-binding motifs within

*Figure 4 continued on next page*

*Figure 4 continued*

*Sel1l* or *Hrd1* promoter luciferase reporters (as in **L**, **M**) are shown. (**F–I**). Upper: Schematic diagram of the ChIP primer (P1–P3) locations across the *Sel1l* (**F**) *Hrd1*, (**G**) *Edem1*, (**H**) or *Os9* (**I**) promoter regions. TSS: transcription start site. Lower: Chromatin extracts from EL4 cells treated with PBS or 5 μg/ml DLL4 for 24 hr were subjected to ChIP using anti-RBP-J antibody, anti-NICD antibody, or normal IgG. Genomic regions of *Sel1l* (**F**), *Hrd1* (**G**), *Edem1* (**H**), or *Os9* (**I**) promoter (as in left panel) were tested for enrichment of RBP-J, NICD or IgG. Data are shown as percentage of input. (**J**) *Sel1l* or *Hrd1* promoter luciferase reporter was co-transfected with empty vector or different doses of NICD into HEK293T cells, and luciferase activity was measured 36 hr after transfection. pGL3 basic was used as control. (**K and L**) Wild-type or mutant (RBP-J motif mutations, as shown in **E**) *Sel1l* (**K**) or *Hrd1* (**L**) promoter luciferase reporter was transfected into EL4 cells which were treated with PBS or 5 μg/ml DLL4 for 24 hr before harvest. Luciferase activity was measured 36 hr after transfection. All luciferase data are presented relative to *Renilla* readings. Data are shown as mean ± s.d. Two-tailed Student's t-tests (**A, B, D, F-I, K, L**) or one-way ANOVA with Bonferroni test (**J**) were used to calculate p values. n.s., not significant, \*p < 0.05, \*\*p < 0.01, \*\*\*p < 0.001, \*\*\*\*p < 0.0001.

The online version of this article includes the following source data and figure supplement(s) for figure 4:

**Source data 1.** Original western blot images shown in *Figure 4*.
**Source data 2.** Excel file containing numerical values shown in *Figure 4*.
**Figure supplement 1.** Notch signal regulates ERAD genes expression.
**Figure supplement 1—source data 1.** Excel file containing numerical values shown in *Figure 4—figure supplement 1*.

To further evaluate whether the *Sel1l* CKO mice phenotype was due to defective pre-TCR signaling, we introduced the MHCII-restricted TCR-β transgene OT-II into *Sel1l* CKO mice. As previously reported (*Kim et al., 2014*; *Marquis et al., 2014*), expression of TCR transgenes like OT-II in early DN thymocytes can rescue β-selection defects caused by defective pre-TCR expression or signaling. However, the OT-II TCR transgene did not rescue T cell development in OT-II. *Sel1l* CKO mice which showed a >80% decrease in thymic cellularity, DP and SP thymocytes (*Figure 5—figure supplement 1E–G*). In addition, OT-II TCR failed to rescue impaired DN3 to DN4 thymocyte transition resulting from *Sel1l* deficiency as DN4 thymocytes were decreased ~60% in OT-II. *Sel1l* CKO mice compared to littermate controls (*Figure 5—figure supplement 1H*). Taken together, these data implied that β-selection defects in *Sel1l* CKO mice were not due to defects in pre-TCR expression or signaling.

### *Sel1l*-deficiency triggers unresolved ER stress during β-selection

To understand the molecular mechanism by which SEL1L ERAD regulates thymocyte survival and differentiation at β-selection, we performed RNA-seq on control and *Sel1l* CKO DN3 thymocytes. The most upregulated pathways in *Sel1l* CKO DN3 thymocytes were ER stress response and the UPR, including both IRE1α and PERK pathways (*Figure 5A,B*). Gene set enrichment analysis (GSEA) also revealed enriched ER stress response in *Sel1l* CKO thymocytes (*Figure 5C–E*). These signatures hinted at elevated ER stress in *Sel1l* CKO DN3 thymocytes. Consistent with elevated ER stress, flow cytometry quantification of ER tracker dye staining indicated significant ER expansion in *Sel1l* CKO DN3 cells compared to control thymocytes (*Figure 5F,G*).

To further ascertain the induction of ER stress following *Sel1l* deletion, we sorted DN3 and DN4 thymocytes from control or *Sel1l* CKO mice and examined the activation of all three UPR branches. We found a significant increase in IRE1α proteins and in the splicing of its substrate *Xbp1* in *Sel1l* CKO thymocytes (*Figure 5H,I* and *Figure 5—figure supplement 2A,B*). We also observed increased PERK and eIF2α phosphorylation as well as increased ATF4 and BIP proteins in *Sel1l* CKO DN3 thymocytes (*Figure 5H*). Various ER chaperones, including *Calreticulin*, *Grp94 (Hsp90b1)*, *Hspa5(Bip)*, *Hyou1*, and *Canx*, and other members of ERAD pathway were markedly upregulated in *Sel1l* CKO DN3 and DN4 thymocytes (*Figure 5—figure supplement 2C,D*). These data indicate that *Sel1l* deletion triggers ER stress leading to activation of all three UPR branches in DN thymocytes.

As ERAD alleviates proteotoxic stress by promoting the degradation of misfolded or unfolded proteins, we hypothesized that loss of SEL1L increased proteotoxic stress during β-selection. Indeed, we found that *Sel1l* CKO DN3 thymocytes exhibited significantly higher staining for misfolded/unfolded proteins with TMI (*Figure 5J,K*). We also employed proteostat, a molecular rotor dye, to examine protein aggregation in DN3 thymocytes. The proteostat dye specifically intercalates into the cross-beta spine of quaternary protein structures typically found in misfolded and aggregated proteins, which inhibits the dye's rotation and leads to a strong fluorescence (*Figure 5L*). *Sel1l* CKO DN3 thymocytes displayed more protein aggregates compared with control thymocytes (*Figure 5M*,

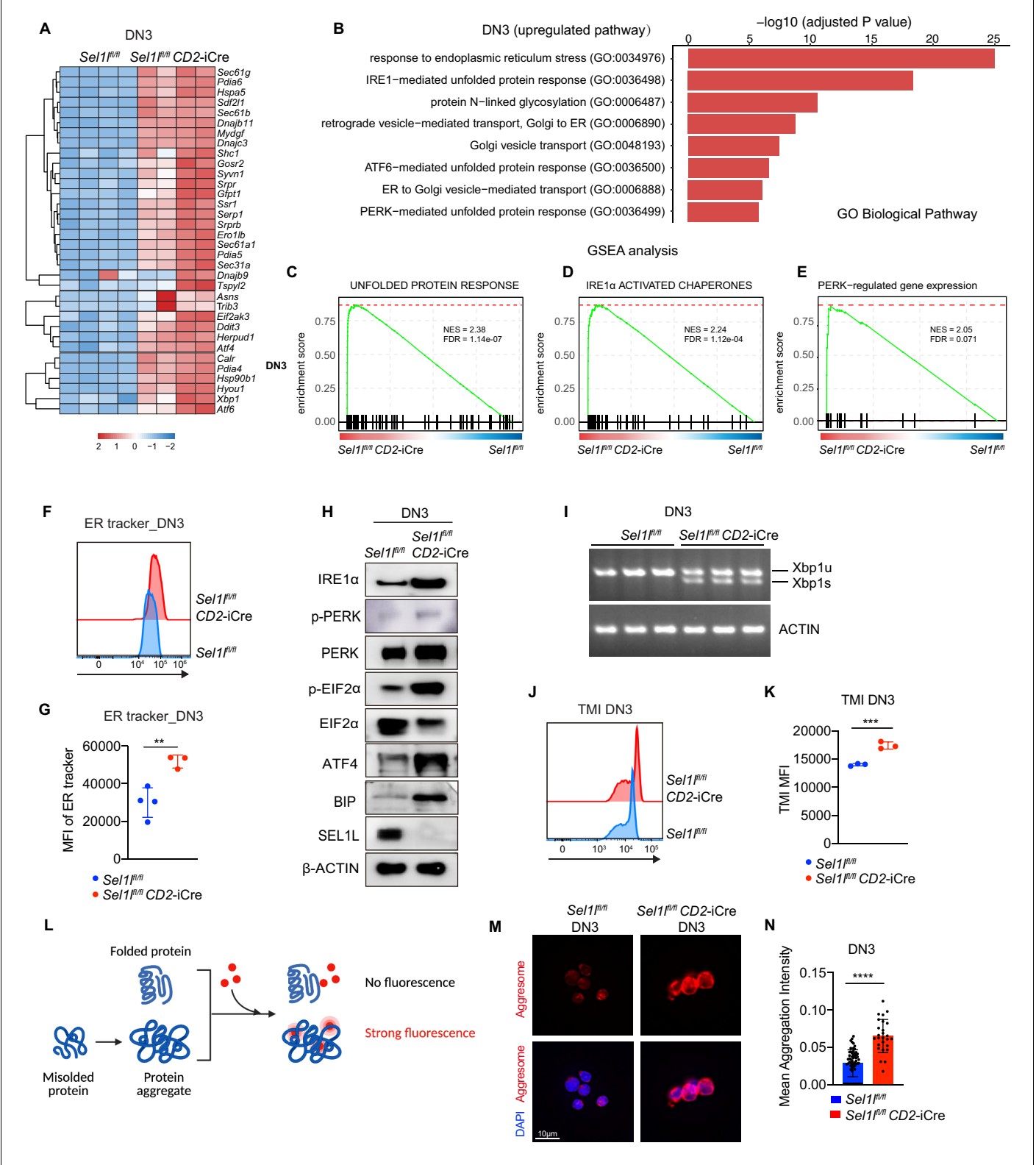

**Figure 5.** *Sel1l*-deficiency triggers unresolved ER stress during β-selection. (**A**) Heatmap showing differentially expressed genes from the RNA-seq analysis of DN3 thymocytes sorted from control (Ctrl, *Sel1l*^flox/flox^) or *Sel1l* CKO (*Sel1l*^flox/flox^; *CD2*-iCre) mice. *n* = 4. (**B**) Gene Ontology (GO) analysis of the most significantly upregulated pathways in *Sel1l* CKO (*Sel1l*^flox/flox^; *CD2*-iCre) DN3 thymocytes compared with control (Ctrl, *Sel1l*^flox/flox^) DN3 thymocytes. (**C–E**) Plots from GSEA analysis showing enrichment of Unfolded Protein Response (**C**), IRE1α (**D**), and PERK (**E**) pathways in *Sel1l* CKO

*Figure 5 continued on next page*

Figure 5 continued

(*Sel1l*$^{flox/flox}$; *CD2*-iCre) DN3 thymocytes compared to control (Ctrl, *Sel1l*$^{flox/flox}$) DN3 thymocytes. (**F and G**) Representative histogram (**F**) and quantification(**G**) of ER-tracker staining in DN3 thymocytes sorted from control (Ctrl, *Sel1l*$^{flox/flox}$) and *Sel1l* CKO (*Sel1l*$^{flox/flox}$; *CD2*-iCre) mice. Ctrl: n = 4; *Sel1l* CKO: n = 3. MFI, mean fluorescence intensity. (**H**) Western blot analysis of UPR pathway markers in primary DN3 thymocytes sorted from 6-week-old Ctrl or *Sel1l* CKO mice. β-ACTIN was used as loading control. The original western blot images are provided in *Figure 5—source data 1*. (**I**) PCR analysis of XBP1-splicing in DN3 thymocytes sorted from Ctrl or *Sel1l* CKO mice. Xbp1u: Unspliced Xbp1; Xbp1s: Spliced Xbp1. ACTIN was used as loading control. The original gel images are provided in *Figure 5—source data 2*. (**J and K**) Representative histogram (**J**) and quantification (**K**) of unfolded/misfolded protein level measured by TMI in DN3 thymocytes sorted from Ctrl or *Sel1l* CKO mice. n = 3. (**L**) Schematic illustration of labeling and detection of misfolded and aggregated proteins with ProteoStat dye. (**M and N**) Representative images (**M**) and quantification (**N**) of protein aggregation measured by ProteoStat Protein Aggregation Detection Kit in primary DN3 thymocytes sorted from three pooled Ctrl or *Sel1l* CKO mice. Results are shown as mean ± s.d. Two-tailed Student's t-tests (**G, K, N**) was used to calculate p values. **p < 0.01, ***p < 0.001, ****p < 0.0001. The online version of this article includes the following source data and figure supplement(s) for figure 5:

**Source data 1.** Original western blot images shown in *Figure 5*.
**Source data 2.** Original gels images shown in *Figure 5*.
**Source data 3.** Excel file containing numerical values shown in *Figure 5*.
**Figure supplement 1.** SEL1L is not required for TCRβ gene rearrangement and pre-TCR signaling.
**Figure supplement 1—source data 1.** Original gels images shown in *Figure 5—figure supplement 1*.
**Figure supplement 1—source data 2.** Original western blot images shown in *Figure 5—figure supplement 1*.
**Figure supplement 1—source data 3.** Excel file containing numerical values shown in *Figure 5—figure supplement 1*.
**Figure supplement 2.** *Sel1l* knockout induces ER stress.
**Figure supplement 2—source data 1.** Original western blot images shown in *Figure 5—figure supplement 2*.
**Figure supplement 2—source data 2.** Original gels images shown in *Figure 5—figure supplement 2*.
**Figure supplement 2—source data 3.** Excel file containing numerical values shown in *Figure 5—figure supplement 2*.

**N**). Collectively, these results corroborate that DN thymocytes lacking SEL1L accumulate misfolded/unfolded proteins leading to proteotoxic stress that then trigged the UPR and eventually apoptosis.

## PERK signaling drives β-selected thymocyte apoptosis in *Sel1l* CKO mouse

To determine whether upregulation of the UPR contributed to post-β-selected thymocyte apoptosis and the *Sel1l* CKO mouse phenotype, we generated *Sel1l/Xbp1* double-knockout (*Sel1*$^{flox/flox}$.*Xbp1*$^{flox/flox}$.*CD2*-iCre), and *Sel1l/Perk* double-knockout (*Sel1*$^{flox/flox}$.*Perk*$^{flox/flox}$.*CD2*-iCre) mice. Deletion of *Xbp1* in *Sel1l* CKO mice did not rescue thymocyte development (*Figure 6—figure supplement 1A*). In fact, *Sel1l/Xbp1* double-knockout (DKO) mice showed more severe thymocytes development defects including a more than 95% loss in thymus cellularity and DP thymocyte numbers (*Figure 6—figure supplement 1A*). The DN3 and DN4 thymocytes from *Sel1l/Xbp1* DKO exhibited much more *Chop* expression (*Figure 6—figure supplement 1B*). These data suggest that induction of the IRE1α/XBP1 pathway functions as a compensatory adaptive pathway to restrain *Sel1l*-deficiency induced ER stress.

In contrast to deletion of *Xbp1*, we found that deletion of *Perk* significantly rescued the *Sel1l* CKO mouse phenotype evident in the near complete restoration of thymus cellularity, DP and SP thymocyte cell numbers (*Figure 6A,B*). *Perk* deletion also restored peripheral T cells in spleen and lymph nodes compared to *Sel1l* CKO mice (*Figure 6C–F*). Consistent with the rescue, *Perk* deletion significantly reduced *Sel1l*-deficiency induced *Chop* induction (*Figure 6G*) and thymocyte apoptosis (*Figure 6H,I*) and restored normal cell cycle kinetics to *Sel1l* CKO DN4 thymocytes (*Figure 6—figure supplement 1C*). As *Perk* deficiency alone had no effect on T cell development (*Figure 6A–I*), these results indicate that activated PERK signaling contributed to the apoptosis of DN3/DN4 thymocytes that impaired β-selection in *Sel1l* CKO mice. Collectively, we conclude that SEL1L-ERAD promotes β-selected DN thymocyte differentiation by maintaining ER proteostasis and suppressing ER stress-induced cell death through the PERK pathway.

## Discussion

In this study, we have uncovered a novel 'Notch-ERAD' axis in thymocyte development. In the absence of the core ERAD protein SEL1L, thymocytes failed to survive β-selection and T cell

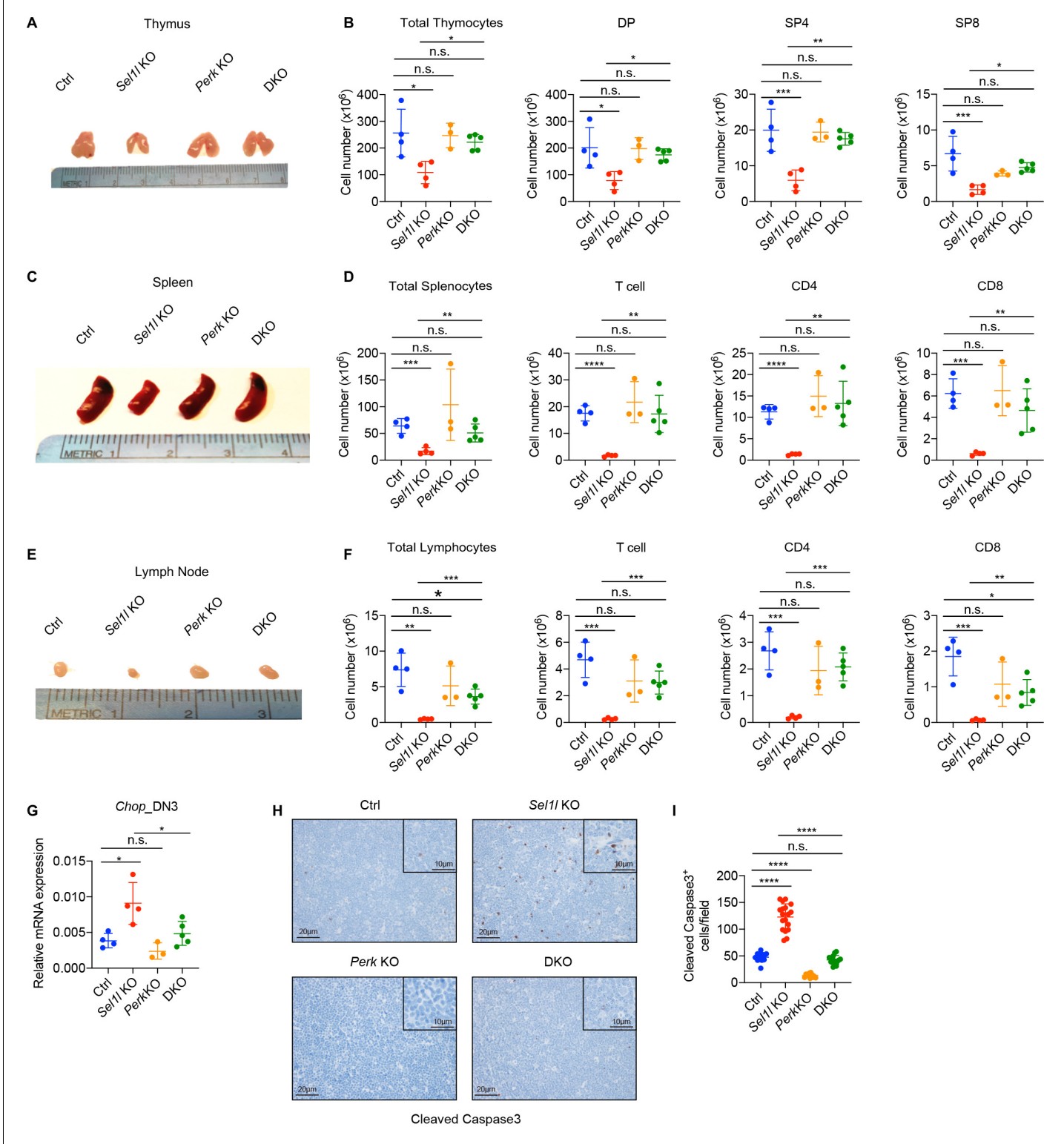

**Figure 6.** PERK signaling drives β-selected thymocyte apoptosis in *Sel1l* CKO mouse. (**A and B**) Representative images of thymus (**A**) and quantification of total thymocytes, DP, SP4 and SP8 thymocytes (**B**) from age (6-week-old) and gender-matched control (Ctrl, *Sel1l*$^{flox/flox}$), *Sel1l* KO (*Sel1l*$^{flox/flox}$; *CD2*-iCre), *Perk* KO (*Perk*$^{flox/flox}$; *CD2*-iCre)and *Sel1l/Perk* double knockout (DKO. *Sel1l*$^{flox/flox}$, *Perk*$^{flox/flox}$; *CD2*-iCre) mice. $n$ = 3–5 each group. (**C and D**) Representative images of spleen (**C**) and quantification of total splenocytes, total CD3$^+$ T cells, CD4$^+$ T cells, and CD8$^+$ T cells (**D**) from the same mice with indicated genotype as in **A** and **B**. $n$ = 3–5 each group. (**E and F**) Representative images of the inguinal (left) lymph node (**E**) and quantification of total lymphocytes, total CD3$^+$ T cells, CD4$^+$ T cells, and CD8$^+$ T cells (**F**) from the same mice with indicated genotype as in **A** and **B**. $n$ = 3–5 each

*Figure 6 continued on next page*

*Figure 6 continued*

group. (**G**) Quantitative RT–PCR analysis of *Chop* expression in DN3 thymocytes sorted from mice with indicated genotype. *n* = 3–5 each group. (**H and I**) Representative images (**H**) and quantification (**I**) of cleaved caspase-3 (CC3)-positive cells in the thymus of 6- to 8-week-old gender-matched mice with indicated genotype. Twelve fields were counted at ×20 magnification from four mice with indicated genotype. Scale bars are indicated. Data are representative of three independent experiments and are shown as mean ± s.d. The statistical significance was calculated by two-tailed unpaired t-test (**D, F**) One-way ANOVA with turkey test (**B, G**) or one-way ANOVA with Bonferroni test (**I**). ns, not significant, *p < 0.05, **p < 0.01, ***p < 0.001, ****p < 0.0001.

The online version of this article includes the following source data and figure supplement(s) for figure 6:

**Source data 1.** Excel file containing numerical values shown in *Figure 6*.
**Figure supplement 1.** XBP1 functions as a compensatory adaptative mechanism in *Sel1l* CKO mouse.
**Figure supplement 1—source data 1.** Excel file containing numerical values shown in *Figure 6—figure supplement 1*.

development was severely impaired. Our results imply that the protein synthesis and folding demands during β-selection require a robust proteome quality control monitoring which is accomplished by the ERAD in thymocytes transitioning from the DN3 to DN4 stage. Thus, induction of the SEL1L axis of ERAD, which peaks at the DN3 stage, represents a previously undefined and critical ER proteostasis checkpoint during β-selection. It is notable that this ER proteostasis checkpoint is also regulated by the same Notch signals which, together with the pre-TCR, induce β-selection in DN3 thymocytes. We identified that Notch1 and RBP-J directly bind to most ERAD gene promoters and directly regulate their expression. In fact, we observed that Notch-signal induced SEL1L protein levels exceeded the change in *Sel1l* mRNA expression (*Figure 4B,C*), suggesting additional post-transcriptional regulation of ERAD by Notch that warrants further investigation.

ERAD and UPR are two key ER protein quality control machineries that are activated at different thresholds. ERAD is responsible for the clearance of misfolded proteins at steady-state and is constitutively activated regardless of ER stress. In contrast, the UPR is a stress response pathway that is triggered when the accumulation of misfolded and unfolded proteins exceed the ER folding capacity. Existing evidence suggest that UPR pathways like the IRE1α-XBP1 axis appear to selectively regulate hematopoietic cell differentiation programs but individual UPR enzymes influence early thymopoiesis remain to be fully characterized. For instance, the IRE1α-XBP1 is essential for eosinophil (*Bettigole et al., 2015*), dendritic cell (*Cubillos-Ruiz et al., 2015*; *Iwakoshi et al., 2007*; *Osorio et al., 2014*), NK (*Dong et al., 2019*; *Wang et al., 2019*), and plasma cell (*Reimold et al., 2001*; *Shapiro-Shelef and Calame, 2005*) differentiation. Unlike the marked defect in thymocyte development in the absence of SEL1L-ERAD, our current study now clearly demonstrates that individual UPR regulators are dispensable for β-selection and subsequent stages of thymocyte maturation. We specifically found that deletion of the individual UPR master regulators XBP1, ATF6, or PERK had no overt impact on αβ T cell development in the thymus and peripheral tissues. Nevertheless, our results don't exclude the possibility that redundancy might exist among the UPR pathways during thymocyte development.

*Sel1l*-deficiency impaired the survival of DN thymocytes following β selection and resulted in the inability of DN3 thymocytes to expand and progress to DP thymocytes. Interestingly, this phenotype was not due to defective rearrangement of TCRβ nor defective Notch or pre-TCR signaling which both activate pro-survival genes in β-selected thymocytes (*Ciofani and Zúñiga-Pflücker, 2005*; *Kreslavsky et al., 2012*; *Zhao et al., 2019*). Instead, our genetic inactivation delineate a pro-apoptotic pathway driven by the PERK axis. While all three UPR pathways can promote apoptosis and appear to be upregulated in *Sel1l* CKO thymocytes, it is notable that only PERK axis induces apoptosis in *Sel1l* CKO thymocytes. On the contrary, induction of the IRE1α-XBP1 axis of the UPR can help cells adapt to stress. Consistent with this view, deletion of *Xbp1* in *Sel1l* CKO thymocytes exacerbated defects in thymocyte development unlike PERK inactivation which restored T cell development from *Sel1l* CKO thymocytes.

Although αβ T and γδ T cells develop in the thymus from the same thymic seeding progenitors, the function of ERAD appears to be restricted to αβ T cells. While future studies are needed to determine the alternative protein quality control mechanism regulating γδ T cell development, our findings are consistent with the selective requirement for Notch in driving β-selection (*Maillard et al., 2006*), a key developmental checkpoint unique to the αβ T cell differentiation program. In addition, DN thymocytes undergoing β selection expand more than 100-fold, a situation

that likely explains their increased translational activity. That post-β-selected DN thymocytes displayed markedly low levels of misfolded/unfolded proteins support the view that upregulation of the SEL1L-ERAD axis in DN3 thymocytes provides a mechanism to alleviate deleterious proteotoxic stress that compromise the fitness of DN4 thymocytes. In this way, SEL1L-ERAD safeguards to DN3/DN4 thymocyte pool to ensure that the maximum number of DN thymocytes expressing a functional TCRβ protein progress to the DP stage to audition for positive selection.

Activating *NOTCH1* mutations are found in more than 50% of human T cell acute lymphoblastic leukemia (T-ALL) patients (*Weng et al., 2004*) wherein Notch plays pivotal roles in regulating the survival and metabolism of T-ALL cells (*Aster et al., 2011*; *García-Peydró et al., 2018*). It will be of interest in future studies to investigate whether Notch-regulated ERAD also promotes ER proteostasis and sustains T-ALL survival by clearing misfolded proteins and restricting ER stress-induced cell death.

In summary, our study reports a previously unknown function of Notch in maintaining proteostasis and protecting post-β-selection thymocytes from ER stress by upregulating ERAD. We propose that in addition to driving energy metabolism and survival programs demanded by proliferating thymocytes following β-selection, Notch signals in parallel activate the ERAD machinery in DN3 thymocytes to clear misfolded proteins. Thus, stringent protein quality control through the SEL1L-ERAD pathway is required for successful β-selection and the development of the αβ T cells that mediate adaptive immunity.

## Materials and methods

Key resources table is in Appendix 1.

### Mice

The mice were maintained in a pure C57BL/6 background and kept under specific-pathogen-free conditions in the transgenic mouse facility of the Baylor College of Medicine (22–24°C, 30–70% humidity with a 12 hr dark and 12 hr light cycle). *Sel1l*$^{flox/flox}$ and *Xbp1*$^{flox/flox}$ were described previously (*Xu et al., 2020*), *Perk*$^{flox/flox}$ mice were purchased from Jackson Laboratory (Stock No. 023066). The floxed mice were crossed with either h*CD2*-iCre (The Jackson Laboratory, 008520) or *Cd4-Cre* (The Jackson Laboratory, 022071) mice to generate *Sel1l*$^{flox/flox}$; *CD2*-iCre, *Sel1l*$^{flox/flox}$; *Cd4-Cre, Xbp1*$^{flox/flox}$; *CD2*-iCre, or *Perk*$^{flox/flox}$; *CD2*-iCre mice. The *Sel1l*$^{flox/flox}$; *Xbp1*$^{flox/flox}$; *CD2*-iCre mice and *Sel1l*$^{flox/flox}$; *Perk*$^{flox/flox}$; *CD2*-iCre mice were generated by crossing *Sel1l*$^{flox/flox}$; *CD2*-iCre mice with *Xbp1*$^{flox/flox}$ or *Perk*$^{flox/flox}$ mice respectively. *Atf6*$^{flox/flox}$; *Vav1*-iCre mice were generated by crossing *Atf6l*$^{flox/flox}$ (The Jackson Laboratory, 028253) with *Vav1*-iCre mice (The Jackson Laboratory, 008610). *Sel1l* CKO; OTII transgenic mice were generated by crossing *Sel1l* CKO mice with OTII transgenic mice (The Jackson Laboratory, 004194).

Six- to 8-week-old gender-matched mice were used for phenotype analysis and in vitro assay. For the bone marrow transplantation assays, female C57BL/6-Ly5.1 (CD45.1$^+$) mice (Charles River, 564) were used at 8–12 weeks of age. All procedures were approved by the Baylor College of Medicine Institutional Animal Care and Use Committee or Case Western Reserve University Institutional Animal Care and Use Committee. The study is compliant with all of the relevant ethical regulations regarding animal research.

### In vivo assays

For the competitive bone marrow transplantation experiments, CD45.1$^+$-recipient mice were lethally irradiated (10 Gy, delivered with two equal doses 4 hr apart) and injected retro-orbitally with 1x10$^6$ whole bone marrow cells from *Sel1l*$^{flox/flox}$ or *Sel1l*$^{flox/flox}$; *CD2*-iCre mice with an equal number of CD45.1$^+$ competitor BM cells. For the BrdU incorporation assay, the mice were injected intraperitoneally with single dose of BrdU (BD, 559619; 50 mg/kg body weight) 2 hr before euthanization.

### Flow cytometry and cell sorting

Single-cell suspensions from thymus, spleen and lymph nodes were obtained by passing the tissues through a 70 μm strainer. Bone marrow was removed from femurs and tibiae by flushing with PBS, 2% FBS. For bone marrow and spleen cell suspensions, red blood cells were removed by incubating with RBC Lysis Buffer (Biolegend, 420301). $1 \times 10^6$ to $2 \times 10^6$ cells were stained with antibodies at

4°C for 30 min in phosphate-buffered saline (PBS) containing 1% bovine serum albumin (BSA), followed by washing to remove the unbound antibodies.

Antibodies used for flow cytometry: The biotin-conjugated lineage markers for excluding non-DN cells are purchased from BioLegend: CD11b (M1/70, 101204), CD11c (N418, 117303), Ter119 (Ter119, 116204), Gr-1 (RB6-8C5, 108404), CD49b (DX5, 108904), and B220 (RA3-6B2, 103204). CD4 (BV650, RM4-5, 100546, BioLegend), CD8 (AF700, 53–6.7, 100730, BioLegend), CD25 (PE-Cy7, 3C7, 101915, BioLegend), CD44 (PE594, IM7, 103056, BioLegend), c-Kit (APC-Cy7, 2B8, 105838, Biolegend), γδTCR (PE, GL3, 118108, Biolegend), and CD27 (APC, LG.3A10, 124211, BioLegend) antibodies were used for analysis of ETP to SP cells in the thymus. CD4 (BV650, RM4-5, 100546, BioLegend), CD8 (AF700, 53–6.7, 100730, BioLegend), γδTCR (APC, GL3, 118115, BioLegend), B220 (PB, RA3-6B2, 103230, BioLegend), and NK1.1 (FITC, PK136, 108706, BioLegend) antibodies were used for the analysis of mature cells in the spleen and lymph nodes. Anti-CD45.1 (FITC, A20, 110706, BioLegend) and CD45.2 (APC, 104, 109814, BioLegend) antibodies were used for the analyses of donor chimerism in the bone marrow transplantation assay. All antibodies used in this study are listed in Appendix 1. Dead cells were excluded by 4,6-diamidino-2-phenylindole (DAPI) staining.

For cell sorting, DN thymocytes were purified by negative selection using a magnetic bead/column system (CD4 (L3T4) MicroBeads, 130-117-043; CD8a (Ly-2) MicroBeads, 130-117-044; LD Columns, 130-042-901; all purchased from Miltenyi Biotec) in accordance to the manufacturer's instructions. The pre-enriched DN cells were further stained as described above to sort DN2, DN3, or DN4 cells. Dead cells were excluded by DAPI staining.

Flow cytometry data were collected using BD FACS Diva eight on a BD LSR II or BD Fortessa analyzer. The cell-sorting experiments were performed on a FACS Aria II cell sorter (BD). The acquired data were analyzed using the FlowJo 10 software.

## Apoptosis assay

For Annexin V staining, thymocytes were first stained with surface makers and then stained with Annexin V antibody for 30 min at room temperature in Annexin V binding buffer (TONBO, TNB-5000-L050). The cells were resuspended in Annexin V binding buffer with 1 µg/ml DAPI for analysis.

## Cell proliferation assay

For cell cycle analysis, thymocytes were stained with surface markers followed by fixation and permeabilization with eBioscience Transcription Factor Staining Buffer Set (Life Technologies, 00-5523-00). The cells were then stained with anti-Ki-67 (FITC, 16A8, 652409, Biolegend) and DAPI for cell cycle analysis. For BrdU staining, total thymocytes were stained with surface markers to define each subset, followed by intracellular staining using the BrdU staining kit according to the manufacturer's instructions (BD Biosciences, 559619).

## Cell culture and cell lines

OP9 bone marrow stromal cells expressing the Notch ligand DL-1 (OP9-DL1) were kindly provided by Dr. Juan Carlos Zúñiga-Pflücker (University of Toronto). The OP9-DL1 cells were cultured and maintained in αMEM medium, supplemented with 10% FBS (Gibco), penicillin (100 µg/mL), and streptomycin (100 U/mL) (Invitrogen). For thymocytes co-cultures, the sorted DN2 or DN3 cells were plated onto confluent OP9-DL1 monolayers (70–80% confluent) with addition of 5 ng/ml recombinant murine interleukin-7 (PeproTech) and 5 ng/ml Flt3L (PeproTech). Cells were harvested at different time points, filtered through 40 µm cell strainer, and stained with antibodies including ani-CD45 V450 (Tonbo, 75–0451 U100), anti-CD4 BV650 (Biolegend, 100546), anti-CD8a AF700 (Biolegend, 100730), anti-CD44 FITC (Biolegend, 103006), or anti-CD25 APC (Biolegend, 101910), followed by staining with Annexin V PE (Biolegend, 640908) and DAPI (4,6-diamidino-2-phenylindole, Invitrogen, D1306). EL4 cells (ATCC-TIB39) and HEK293T cells (ATCC CRL-3216) were cultured in DMEM with 10% FBS (Gibco), penicillin (100 µg/mL) and streptomycin (100 U/mL) (Invitrogen). No cell lines were listed in the database of cross-contaminated or misidentified cell lines by International Cell Line Authentication Committee (ICLAC). Cell lines from ATCC were authenticated by the STR profiling method and tested as mycoplasma contamination free by ATCC.

## Immunohistochemical staining

Thymus was fixed in fresh 4% paraformaldehyde for 24 hr and stored in 70% ethanol until paraffin embedding. Hematoxylin and eosin staining were performed on 5 μm–thick paraffin sections. For cleaved caspase-3 IHC, the anti-cleaved caspase-3 antibody (1:50, Cell Signaling Technology #9661) was used. Slides were incubated with Envision Labeled Polymer-HRP Anti-Rabbit (Dako, K4002) for 30 min. Sections were developed with DAB+ solution (Dako, K3468) and counterstained with Harris Hematoxylin. Imaging analysis was performed with ImageJ (FIJI) to automatically count the labeled cells in each region, the data were shown as number of positive cells per region.

## RNA extraction and quantitative real-time PCR

Thymocytes were sorted directly into TRIzol LS reagent (Invitrogen, 10296010). Total RNA was extracted according to the manufacturer's instructions. Total RNA was reverse-transcribed using High-Capacity cDNA Reverse Transcription Kit (Thermo Fisher Scientific, 4368813). Quantitative real-time PCR was performed using PowerUp SYBR Green Master Mix (Thermo Fisher Scientific, A25778) on QuantStudio six real-time PCR system (Applied Biosystems). The primer sequences are listed in Appendix 1.

## Western blot analysis

Western blot was performed as described previously (*Xu et al., 2020*). Approximately $5 \times 10^5$ DN3 cells were sorted directly into 250 μl PBS containing 20% trichoracetic acid (TCA). The concentration of TCA was adjusted to 10% after sorting and cells were incubated on ice for 30 min before centrifugation at 13,000 rpm for 10 min at 4°C. Precipitates were washed twice with pure acetone (Fisher scientific, A18-4) and solubilized in 9 M urea, 2% Triton X-100, and 1% dithiothreitol (DTT) in 1 x LDS sample buffer (Invitorgen, NP0007). Samples were separated on NuPAGE 4–12% Bis-Tris protein gels (Invitrogen, NP0336BOX) and transferred to PVDF membrane (Millipore). The blots were incubated with primary antibodies overnight at 4°C and then with secondary antibodies. Blots were developed with the SuperSignal West Femto chemiluminescence kit (Thermo Scientific, 34096). Antibodies and reagents used are in Appendix 1. The original western blot images are in source data files.

## RNA-seq and analysis

DN3 thymocytes were directly sorted into TRIzol LS (ThermoFisher, cat. 10296028) and RNA was extracted following the standard protocol. The cDNA libraries were prepared using Truseq Stranded mRNA Kit (Illumina, California, USA # 20020594). Sequencing was performed on Illumina HiSeq 2000 (Illumina, California, USA), 150 bp paired end. Quality control was performed using FastQC. Raw reads were aligned to mouse genome GRC38 using STAR (2.5.2b) and counts for each protein-coding gene were obtained with HTSeq (2.7). DESeq2 package (1.26.0) in R (3.6.1) was used to perform analysis of differential gene expression. Upregulated genes (with adjusted p value < 0.05, log2 fold change > 0.59) were selected for Gene Ontology (GO) analysis using Enrichr (https://maayanlab.cloud/Enrichr/). We further performed Gene Set Enrichment Analysis (GSEA) using fgsea package (1.11.2) with padj-preranked gene lists and mouse gene set collection from Bader Lab. Heatmaps were generated with pheatmap (1.0.12).

## Chromatin immunoprecipitation (ChIP) assay

For *Figure 4C*, cell culture plates were pre-coated with PBS or 5 μg/ml recombinant mouse DLL4 (BioLegend, #776706) overnight at 4°C before seeding EL4 cells. For *Figure 4B,F–I,K and L*, soluble DLL4 (5 μg/ml) was used. EL4 cells were incubated with soluble DLL4 for 24 hr before crosslinked with 1% formaldehyde for 10 min at room temperature. Reaction was quenched with 125 mM glycine. ChIP was performed as previously described (*Zhao et al., 2018*) with NOTCH1 NICD antibody (Abcam, ab27526), RBPJ antibody (Cell Signaling Technology, #5313), or normal rabbit IgG (Cell Signaling Technology, #2729). The sequences of all ChIP primers are listed in Appendix 1.

## Luciferase assay

The firefly luciferase reporter for *Sel1l* or *Hrd1* (*Syvn1*) promoter was constructed by cloning the genomic region into the *Mlu*I and *Xho*I sites or the *Mlu*I and *Hind*III sites in the pGL-3 basic vector

(Promega), respectively. Mutations were made by overlap extension polymerase chain reaction as previously described (*Bryksin and Matsumura, 2010*). All constructs were verified by DNA sequencing. The sequences of all primers are listed in Appendix 1. HEK293T cells were transfected with *Sel1l* or *Hrd1* promoter constructs, pRL-PGK (Promega) and 3xFlag-NICD1 (Addgene, #20183) or empty using Lipofectamine 3000 (Invitrogen, L3000015). Cell lysates were collected 48 hr after transfection, and luciferase activities were analyzed using the dual-luciferase reporter assay system (E1910, Promega). pRL-PGK, which expresses Renilla luciferase, was used as the internal control for adjustment of discrepancies in transfection and harvest efficiencies. EL4 cells were transfected with *Sel1l* or *Hrd1* promoter constructs and pRL-PGK (Promega) using Lipofectamine 3000 (Invitrogen, L3000015). Cells were incubated with PBS or 5 µg/ml recombinant mouse DLL4 (BioLegend, #776706) for 24 hr before analysis.

## V(D)J recombination assay

A total of $5 \times 10^5$ DN3 thymocytes were sorted from control (Ctrl, *Sel1l^{flox/flox}*) or *Sel1l* CKO (*Sel1l-^{flox/flox}*; *CD2*-iCre) thymus and subjected to genomic DNA isolation using PureLink Genomic DNA Mini Kit (K1820-02) according to the manufacturer's instructions. The amplification of eF-1 fragment was used as input control. The primers used for Vβ5-Jβ2, Vβ8- Jβ2, Vβ11-Jβ2, and eF1 are listed inAppendix 1. PCR products were resolved by 2% agarose gel electrophoresis.

## ER tracker staining

Sorted DN3 thymocytes were washed with PBS, incubated with 1 µM ER-Tracker Green (Thermo Fisher, E34251) in PBS for 15 min at 37°C. The cells were then washed and resuspended in PBS, and analyzed by flow cytometry.

## Tetraphenylethene maleimide (TMI) staining

$5x10^6$ thymocytes were stained for cell surface markers as described above. After surface markers staining, cells were washed twice with PBS. Tetraphenylethene maleimide (TMI; 2 mM in DMSO) was diluted in PBS to reach 50 µM final concentration and stained samples for 30 min at 37°C. Samples were washed once with PBS and analyzed by flow cytometry.

## In vivo measurement of protein synthesis

O-propargyl-puromycin (OPP; MedChem Source LLP) stock dissolved in 10% DMSO/PBS was diluted in PBS for intraperitoneal injection at 50 mg/kg. Mice were weighed, individually injected with OPP or vehicle, and euthanized 1 hr after injection. Thymuses were immediately harvested. Thymocytes were isolated and stained for surface antigens and viability dyes, which was followed by fixation and permeabilization according to the Click-iT Plus OPP Alexa Fluor 647 Protein Synthesis Assay Kit (ThermoFisher Scientific). Briefly, after incubation cells were subsequently fixed in 4% paraformaldehyde, permeabilized with 0.5% Triton-X100, then incubated with the AF647 reaction cocktail. Samples were acquired using a BD LSRFortessa and analyzed using FlowJo (Becton Dickinson) as per flow cytometry methods.

## Protein aggregation detection assay

The PROTEOSTAT Aggresome Detection kit (Enzo Life Sciences, ENZ-51035–0025) was used to detect protein aggregates in freshly sorted DN3 thymocytes according to the manufacturer's instructions. DN3 thymocytes were fixed, permeabilized and incubated with PROTEOSTAT dye (1:10,000 dilution) for 30 min at room temperature. Nuclei were counterstained with DAPI. Samples stained with DAPI only were used as negative controls. Images (16-bit greyscale TIFFs) were analyzed using CellProfiler v2.2. In brief, the DAPI channel images were first smoother with a median filter and nuclei identified with automatic thresholding and a fixed diameter. Nuclei touching the border of the image are eliminated. Touching nuclei are separated with a watershed algorithm. Then, cell boundaries were identified by watershed gradient based on the dye signal, using nuclei as a seed. Metrics were extracted from the cell, cytoplasm, and nuclear compartments.

## TUNEL staining

TUNEL staining was performed on paraffin-embedded tissue sections using the In Situ Cell Death Detection Kit (catalog 11684795910, Roche) following the manufacturer's instructions. Sections were counterstained with DAPI, and images were captured under fluorescence microscope. Tissue sections incubated with TUNEL reaction buffer without dTd enzyme served as negative controls. Tissue sections treated with DNase I served as positive controls. The quantification of TUNEL$^+$ cells was performed with ImageJ (FIJI) to automatically count the labeled cells in each region. The data were presented as number of positive cells per region.

## Statistics and reproducibility

Data are expressed as the mean ± s.d. or mean ± s.e.m. as indicated in the figure legends; $n$ is the number of independent biological replicates, unless specifically indicated otherwise in the figure legend. The respective $n$ values are shown in the figure legends. The mice used for bone marrow transplantation were randomized and no blinding protocol was used. No statistical method was used to pre-determine the sample sizes. The results were quantified using GraphPad Prism 8. Student's t test was utilized to compare the differences between two groups. One-way ANOVA with Tukey's or Bonferroni's multiple comparison test was used to compare the differences among three or more groups. Two-way ANOVA with Bonferroni's post test was used to calculate the significance for in vitro DN2 differentiation measurement over time.

## Study approval

All protocols described in this study were approved by the Baylor College of Medicine Institutional Animal Care and Use Committee (protocol: AN-6813) or Case Western Reserve University Institutional Animal Care and Use Committee (protocol: 2017–0055).

## Acknowledgements

We thank Dr. Juan Carlos Zúñiga-Pflücker (University of Toronto) for providing the OP9-DL1 cells and Dr. Laurie Glimcher (Dana Farber Cancer Institute) for providing the *Xbp1* flox mice. This work was supported by the Cytometry and Cell Sorting Core at the Baylor College of Medicine with funding from the CPRIT Core Facility Support Award (CPRIT-RP180672), the NIH (CA125123, S10OD025251 and RR024574) and the assistance of J M Sederstrom. Imaging for this work was supported by the Integrated Microscopy Core at Baylor College of Medicine and the Center for Advanced Microscopy and Image Informatics (CAMII) with funding from NIH (DK56338, CA125123, ES030285), and CPRIT (RP150578, RP170719), the Dan L Duncan Comprehensive Cancer Center, and the John S Dunn Gulf Coast Consortium for Chemical Genomics.

## Additional information

### Funding

| Funder | Grant reference number | Author |
| --- | --- | --- |
| National Heart, Lung, and Blood Institute | R01HL146642 | Xi Chen |
| National Institute of Allergy and Infectious Diseases | R01 AI1143992 | Stanley Adoro |
| National Cancer Institute | R37CA228304 | Xi Chen |
| National Cancer Institute | K22CA218467 | Stanley Adoro |
| National Cancer Institute | P50CA186784 | Xi Chen |
| National Institute of General Medical Sciences | R35GM130292 | Ling Qi |
| DOD Peer Reviewed Cancer Research Program | W81XWH1910524 | Xi Chen |
| DOD Peer Reviewed Cancer | W81XWH1910306 | Stanley Adoro |

| | | | |
|---|---|---|---|
| Research Program | | | |
| Congressionally Directed Medical Research Programs | W81XWH1910035 | Xiangdong Lv | |
| Cancer Prevention and Research Institute of Texas | RP160283 | Fanglue Peng | |

The funders had no role in study design, data collection and interpretation, or the decision to submit the work for publication.

## Author contributions

Xia Liu, Longyong Xu, Katharine Umphred-Wilson, Data curation, Formal analysis, Validation, Methodology; Jingjing Yu, Data curation, Formal analysis, Validation, Investigation, Methodology, Writing - review and editing; Fanglue Peng, Data curation, Formal analysis, Validation, Investigation, Methodology; Yao Ding, Brendan M Barton, Xiangdong Lv, Michael Y Zhao, Data curation, Formal analysis; Shengyi Sun, Yuning Hong, Ling Qi, Resources; Stanley Adoro, Conceptualization, Formal analysis, Supervision, Validation, Investigation, Writing - original draft, Project administration, Writing - review and editing; Xi Chen, Conceptualization, Formal analysis, Supervision, Funding acquisition, Investigation, Writing - original draft, Project administration, Writing - review and editing

## Author ORCIDs

Jingjing Yu https://orcid.org/0000-0002-2408-5809
Longyong Xu https://orcid.org/0000-0002-8574-4062
Katharine Umphred-Wilson https://orcid.org/0000-0001-6416-0466
Xi Chen https://orcid.org/0000-0002-7995-6202

## Ethics

Animal experimentation: All protocols described in this study were approved by the Baylor College of Medicine Institutional Animal Care and Use Committee (protocol: AN-6813) or Case Western Reserve University Institutional Animal Care and Use Committee (protocol: 2017-0055).

## Decision letter and Author response

Decision letter https://doi.org/10.7554/eLife.69975.sa1
Author response https://doi.org/10.7554/eLife.69975.sa2

# Additional files

## Supplementary files

- Transparent reporting form

## Data availability

Sequencing data have been deposited in GEO under accession code GSE173993. All data generated or analysed during this study are included in the manuscript and supporting files. Source data files have been provided for all Figures.

The following dataset was generated:

| Author(s) | Year | Dataset title | Dataset URL | Database and Identifier |
|---|---|---|---|---|
| Liu X, Yu J, Xu L, Umphred-Wilson K, Peng F, Ding Y, Barton BM, Lv X, Zhao MY, Sun S, Hong Y, Qi L, Adoro S, Chen X | 2021 | Notch-Induced Endoplasmic Reticulum-Associated Degradation Governs Thymocyte Beta-Selection | https://www.ncbi.nlm.nih.gov/geo/query/acc.cgi?acc=GSE173993 | NCBI Gene Expression Omnibus, GSE173993 |

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

# Appendix 1

**Appendix 1—key resources table**

| Reagent type (species) or resource | Designation | Source or reference | Identifiers | Additional information |
|---|---|---|---|---|
| Genetic reagent (*M. musculus*) | *Sel1*$^{flox}$ | PMID:24453213 | | Dr. Ling Qi (Department of Molecular and Integrative Physiology, University of Michigan Medical School) |
| Genetic reagent (*M. musculus*) | *Xbp1*$^{flox}$ | PMID:18556558 | | Dr. Laurie H. Glimcher (Dana Farber Cancer Institute) |
| Genetic reagent (*M. musculus*) | *Perk*$^{flox}$ | Jackson Laboratory | Stock No. 023066 RRID:IMSR_JAX: 023066 | |
| Genetic reagent (*M. musculus*) | h*CD2-iCre* | Jackson Laboratory | Stock No. 008520 RRID:IMSR_JAX: 008520 | |
| Genetic reagent (*M. musculus*) | *Cd4-Cre* | Jackson Laboratory | Stock No. 022071 RRID:IMSR_JAX: 022071 | |
| Genetic reagent (*M. musculus*) | *Atf6l*$^{flox}$ | Jackson Laboratory | Stock No. 028253 RRID:IMSR_JAX: 028253 | |
| Genetic reagent (*M. musculus*) | *Vav1-iCre* | Jackson Laboratory | Stock No. 008610 RRID:IMSR_JAX: 008610 | |
| Genetic reagent (*M. musculus*) | OTII transgenic | Jackson Laboratory | Stock No. 004194 RRID:IMSR_JAX: 004194 | |
| Cell line (*M. musculus*) | EL4 | ATCC | TIB-39 RRID:CVCL_0255 | |
| Cell line (*M. musculus*) | OP9-DL1 | PMID:12479821 | | Dr. Juan Carlos Zúñiga-Pflücker (University of Toronto) |
| Cell line (*Homo sapiens*) | HEK293T | ATCC | CRL-3216 RRID:CVCL_0063 | |
| Antibody | FITC anti-mouse CD45.1, Clone A20 (Mouse monoclonal) | Biolegend | 110706 RRID:AB_313494 | (1:100) FC |
| Antibody | APC anti-mouse CD45.2, Clone 104 (Mouse monoclonal) | Biolegend | 109814 RRID:AB_389211 | (1:100) FC |
| Antibody | Pacific Blue anti-mouse CD45.1, Clone A20 (Mouse monoclonal) | Biolegend | 110721 RRID:AB_492867 | (1:100) FC |
| Antibody | Alexa Fluor 700 anti-mouse CD45.2, Clone 104 (Mouse monoclonal) | Biolegend | 109822 RRID:AB_493731 | (1:100) FC |

*Continued on next page*

*Appendix 1—key resources table continued*

| Reagent type (species) or resource | Designation | Source or reference | Identifiers | Additional information |
|---|---|---|---|---|
| Antibody | FITC anti-mouse/human CD44, Clone IM7 (Rat monoclonal) | Biolegend | 103006 RRID:AB_312957 | (1:100) FC |
| Antibody | PE/Dazzle 594 anti-mouse/human CD44, Clone IM7 (Rat monoclonal) | Biolegend | 103056 RRID:AB_2564044 | (1:100) FC |
| Antibody | APC anti-mouse CD25, Clone 3C7 (Rat monoclonal) | Biolegend | 101910 RRID:AB_2280288 | (1:100) FC |
| Antibody | FITC anti-mouse Ki-67, Clone 16A8 (Rat monoclonal) | Biolegend | 652409 RRID:AB_2562140 | (1:100) FC |
| Antibody | APC/Fire 750 anti-mouse CD117 (c-Kit), Clone 2B8 (Rat monoclonal) | Biolegend | 105838 RRID:AB_2616739 | (1:50) FC |
| Antibody | Brilliant Violet 650 anti-mouse CD4, Clone RM4-5 (Rat monoclonal) | Biolegend | 100546 RRID:AB_2562098 | (1:100) FC |
| Antibody | Alexa Fluor 700 anti-mouse CD8a, Clone 53–6.7 (Rat monoclonal) | Biolegend | 100730 RRID:AB_493703 | (1:100) FC |
| Antibody | PE anti-mouse/rat/human CD27, Clone LG.3A10 (Armenian Hamster monoclonal) | Biolegend | 124209 RRID:AB_1236464 | (1:100) FC |
| Antibody | PE anti-mouse TCR β chain, Clone H57-597 (Armenian Hamster monoclonal) | Biolegend | 109207 RRID:AB_313430 | (1:100) FC |
| Antibody | APC anti-mouse TCR β chain, Clone H57-598 (Armenian Hamster monoclonal) | Biolegend | 109211 RRID:AB_313434 | (1:100) FC |
| Antibody | PE/Cyanine7 anti-mouse CD24, Clone M1/69 (Rat monoclonal) | Biolegend | 101822 RRID:AB_756048 | (1:100) FC |
| Antibody | PE/Dazzle 594 anti-mouse CD3ε, Clone 145–2 C11 (Armenian Hamster monoclonal) | Biolegend | 100348 RRID:AB_2564029 | (1:100) FC |

*Continued on next page*

*Appendix 1—key resources table continued*

| Reagent type (species) or resource | Designation | Source or reference | Identifiers | Additional information |
|---|---|---|---|---|
| Antibody | Pacific Blue anti-mouse/human CD45R/B220, Clone RA3-6B2 (Rat monoclonal) | Biolegend | 103230 RRID:AB_492877 | (1:100) FC |
| Antibody | FITC anti-mouse NK-1.1, Clone PK136 (Mouse monoclonal) | Biolegend | 108706 RRID:AB_313393 | (1:100) FC |
| Antibody | APC Anti-mouse TCR γ/δ, Clone GL3 (Armenian Hamster monoclonal) | Biolegend | 118115 RRID:AB_1731824 | (1:100) FC |
| Antibody | PE Anti-mouse TCR γ/δ, Clone GL3 (Mouse monoclonal) | Biolegend | 118108 RRID:AB_313832 | (1:100) FC |
| Antibody | PE anti-mouse Notch 1, Clone HMN1-12 (Armenian Hamster monoclonal) | Biolegend | 130607 RRID:AB_1227719 | 5 µL/test FC |
| Antibody | violetFluor 450 Anti-Mouse CD45, Clone 30-F11 (Rat monoclonal) | TONBO | 75–0451 U100 RRID:AB_2621947 | (1:100) FC |
| Antibody | PE anti-mouse CD150 (SLAM), Clone TC15-12F12.2 (Rat monoclonal) | BioLegend | 115904 RRID:AB_313683 | (1:100) FC |
| Antibody | TruStain FcX (anti-mouse CD16/32), Clone 93 (Rat monoclonal) | BioLegend | 101320 RRID:AB_1574975 | (1:100) FC |
| Antibody | APC anti-mouse Ly-6A/E (Sca-1), Clone D7 (Rat monoclonal) | BioLegend | 108112 RRID:AB_313349 | (1:100) FC |
| Antibody | PE/Cyanine7 Streptavidin | BioLegend | 405206 | (1:100) FC |
| Antibody | FITC anti-mouse CD48, Clone HM48-1 (Armenian Hamster monoclonal) | BioLegend | 103404 RRID:AB_313019 | (1:100) FC |
| Antibody | Biotin anti-mouse CD3ε, Clone 145–2 C11 (Armenian Hamster monoclonal) | Biolegend | 100304 RRID:AB_312669 | (1:100) FC |

*Continued on next page*

*Appendix 1—key resources table continued*

| Reagent type (species) or resource | Designation | Source or reference | Identifiers | Additional information |
|---|---|---|---|---|
| Antibody | Biotin anti-mouse/ human CD45R/B220, Clone RA3-6B2 (Rat monoclonal) | Biolegend | 103204 RRID:AB_312989 | (1:100) FC |
| Antibody | Biotin anti-mouse TER-119, Clone TER-119 (Rat monoclonal) | Biolegend | 116204 RRID:AB_313705 | (1:100) FC |
| Antibody | Biotin anti-mouse CD49b (pan-NK cells), Clone DX5 (Rat monoclonal) | BioLegend | 108904 RRID:AB_313411 | (1:100) FC |
| Antibody | Biotin anti-mouse/ human CD11b, Clone M1/ 70 (Rat monoclonal) | Biolegend | 101204 RRID:AB_312787 | (1:100) FC |
| Antibody | Biotin anti-mouse CD11c, Clone N418 (Armenian Hamster monoclonal) | Biolegend | 117303 RRID:AB_313772 | (1:100) FC |
| Antibody | Biotin anti-mouse Ly-6G/Ly-6C (Gr-1), Clone RB6-8C5 (Rat monoclonal) | BioLegend | 108404 RRID:AB_313369 | (1:100) FC |
| Antibody | FITC Mouse IgG1, $\kappa$ Isotype Ctrl, Clone MOPC-21 (Mouse monoclonal) | Biolegend | 400108 RRID:AB_326429 | (1:20) FC |
| Antibody | PE Mouse IgG2a, $\kappa$ Isotype Ctrl, Clone MOPC-173 (Mouse monoclonal) | Biolegend | 400213 RRID:AB_2800438 | (1:20) FC |
| Antibody | PERK (D11A8) (Rabbit monoclonal) | Cell Signaling | 5683S RRID:AB_10841299 | (1:200) WB |
| Antibody | Phospho-PERK (Thr980) (16F8) (Rabbit monoclonal) | Cell Signaling | 3179S RRID:AB_2095853 | (1:100) WB |
| Antibody | IRE1$\alpha$ (14C10) (Rabbit monoclonal) | Cell Signaling | 3294S RRID:AB_823545 | (1:200) WB |
| Antibody | eIF2$\alpha$ Antibody (FL-315) (Rabbit polyclonal) | Santa Cruz | sc-11386 RRID:AB_640075 | (1:200) WB |

*Continued on next page*

*Appendix 1—key resources table continued*

| Reagent type (species) or resource | Designation | Source or reference | Identifiers | Additional information |
|---|---|---|---|---|
| Antibody | Phospho-eIF2α (Ser51) (Rabbit polyclonal) | Cell Signaling | 9721S RRID:AB_330951 | (1:100) WB |
| Antibody | Phospho-Lck (Tyr505) (Rabbit polyclonal) | Cell Signaling | 2751 RRID:AB_330446 | (1:100) WB |
| Antibody | Phospho-Zap-70 (Tyr319)/Syk (Tyr352) (65E4) (Rabbit monoclonal) | Cell Signaling | 2717 RRID:AB_2218658 | (1:100) WB |
| Antibody | Zap-70 (D1C10E) XP (Rabbit monoclonal) | Cell Signaling | 3165 RRID:AB_2218656 | (1:200) WB |
| Antibody | CREB-2 (Rabbit polyclonal) | Santa Cruz | sc-200 RRID:AB_2058752 | (1:200) WB |
| Antibody | BiP (C50B12) (Rabbit monoclonal) | Cell Signaling | 3177 RRID:AB_2119845 | (1:200) WB |
| Antibody | Anti-SEL1L (ab78298) (Rabbit polyclonal) | Abcam | ab78298 RRID:AB_2285813 | (1:200) WB |
| Antibody | β-Actin (13E5) (Rabbit monoclonal) | Cell Signaling | 4970S RRID:AB_2223172 | (1:1000) WB |
| Antibody | Anti-Notch 1 (ab27526) (Rabbit polyclonal) | Abcam | ab27526 RRID:AB_471013 | (1:20) ChIP |
| Antibody | RBPSUH (D10A4) XP (Rabbit monoclonal) | Cell Signaling | 5315 RRID:AB_2665555 | (1:50) ChIP |
| Antibody | Normal Rabbit IgG (Rabbit polyclonal) | Cell Signaling | 2729 s RRID:AB_1031062 | (1:250) ChIP |
| Antibody | Cleaved Caspase-3 (Asp175) (Rabbit polyclonal) | Cell Signaling | 9661 RRID:AB_2341188 | (1:50) IHC |
| Sequence-based reagent | *hCD2-iCre* | Jackson Laboratory Stock No. 008520 | 5′ primer | AGATGCCAGGACATCAGGAACCTG |
| Sequence-based reagent | *hCD2-iCre* | Jackson Laboratory Stock No. 008520 | 3′ primer | ATCAGCCACACCAGACACAGAGATC |
| Sequence-based reagent | *Vav1-iCre* | Jackson Laboratory Stock No. 008610 | 5′ primer | AGATGCCAGGACATCAGGAACCTG |
| Sequence-based reagent | *Vav1-iCre* | Jackson Laboratory Stock No. 008610 | 3′ primer | ATCAGCCACACCAGACACAGAGATC |

*Continued on next page*

*Appendix 1—key resources table continued*

| Reagent type (species) or resource | Designation | Source or reference | Identifiers | Additional information |
|---|---|---|---|---|
| Sequence-based reagent | *Sel1l f/f* | PMID:24453213 | 5' primer | TTATGTCTGCTTAATTTCTGCTGG |
| Sequence-based reagent | *Sel1l f/f* | PMID:18556558 | 3' primer | TGAATGAGAAATCCAAGTAGTAGG |
| Sequence-based reagent | *Xbp1 f/f* | PMID:18556558 | 5' primer | ACTTGCACCAACACTTGCCATTTC |
| Sequence-based reagent | *Xbp1 f/f* | PMID:18556558 | 3' primer | CAAGGTGGTTCACTGCCTGTAATG |
| Sequence-based reagent | *Perk f/f* | Jackson Laboratory Stock No. 023066 | 5' primer | TTGCACTCTGGCTTTCACTC |
| Sequence-based reagent | *Perk f/f* | Jackson Laboratory Stock No. 023066 | 3' primer | AGGAGGAAGGTGGAATTTGG |
| Sequence-based reagent | *Atf6 f/f* | Jackson Laboratory Stock No. 028253 | Common Forward | TGCATCTGGGAAGAGAACCA |
| Sequence-based reagent | *Atf6 f/f* | Jackson Laboratory Stock No. 028253 | Wild type Reverse | TGCCATGAACTACCATGTCAC |
| Sequence-based reagent | *Atf6 f/f* | Jackson Laboratory Stock No. 028253 | Mutant Reverse | AGACTGCCTTGGGAAAAGCG |
| Sequence-based reagent | *CD4-iCre* | Jackson Laboratory Stock No. 022071 | Common Forward | GTT CTT TGT ATA TAT TGA ATG TTA GCC |
| Sequence-based reagent | *CD4-iCre* | Jackson Laboratory Stock No. 022071 | Wild type Reverse | TAT GCT CTA AGG ACA AGA ATT GAC A |
| Sequence-based reagent | *CD4-iCre* | Jackson Laboratory Stock No. 022071 | Mutant Reverse | *CTT TGC AGA GGG CTA ACA GC* |
| Sequence-based reagent | OTII transgenic | Jackson Laboratory Stock No. 004194 | Transgene Forward | GCT GCT GCA CAG ACC TAC T |
| Sequence-based reagent | OTII transgenic | Jackson Laboratory Stock No. 004194 | Transgene Reverse | *CAG CTC ACC TAA CAC GAG GA* |
| Sequence-based reagent | Mouse *Sel1l* | this paper | 5' primer | TGAATCACACCAAAGCCCTG |

*Continued on next page*

*Appendix 1—key resources table continued*

| Reagent type (species) or resource | Designation | Source or reference | Identifiers | Additional information |
|---|---|---|---|---|
| Sequence-based reagent | Mouse *Sel1l* | this paper | 3' primer | GCGTAGAGAAAGCCAAGACC |
| Sequence-based reagent | Mouse *Xbp1* | this paper | 5' primer | CTGAGCCCGGAGGAGAAAG |
| Sequence-based reagent | Mouse *Xbp1* | this paper | 3' primer | CTTCCAAATCCACCACTTGC |
| Sequence-based reagent | Mouse *Xbp1s* | this paper | 5' primer | CTGAGTCCGCAGCAGGTG |
| Sequence-based reagent | Mouse *Xbp1s* | this paper | 3' primer | TCCAACTTGTCCAGAATGCC |
| Sequence-based reagent | Mouse *Ddit3(Chop)* | this paper | 5' primer | GTCCCTAGCTTGGCTGACAGA |
| Sequence-based reagent | Mouse *Ddit3(Chop)* | this paper | 3' primer | TGGAGAGCGAGGGCTTTG |
| Sequence-based reagent | Mouse *Erdj4* | this paper | 5' primer | CACAAATTAGCCATGAAGTACC |
| Sequence-based reagent | Mouse *Erdj4* | this paper | 3' primer | TTTCATACGCTTCTGCAATCTC |
| Sequence-based reagent | Mouse *Atf4* | this paper | 5' primer | CCACCATGGCGTATTAGAGG |
| Sequence-based reagent | Mouse *Atf4* | this paper | 3' primer | GTCCGTTACAGCAACACTGC |
| Sequence-based reagent | Mouse *Hrd1* | this paper | 5' primer | CAAGGTCCTGCTGTACATGG |
| Sequence-based reagent | Mouse *Hrd1* | this paper | 3' primer | GTGTTCATGTTGCGGATGGC |
| Sequence-based reagent | Mouse *Atf6* | this paper | 5' primer | AGGGAGAGGTGTCTGTTTCG |
| Sequence-based reagent | Mouse *Atf6* | this paper | 3' primer | CTGCATCAAAGTGCACATCA |
| Sequence-based reagent | Mouse *OS9* | this paper | 5' primer | GGTGTCGGGAGCCTGAATTT |
| Sequence-based reagent | Mouse *OS9* | this paper | 3' primer | CCTCTCTTTCACGTTGGAAGTG |
| Sequence-based reagent | Mouse *Edem1* | this paper | 5' primer | GGGGCATGTTCGTCTTCGG |

*Continued on next page*

*Appendix 1—key resources table continued*

| Reagent type (species) or resource | Designation | Source or reference | Identifiers | Additional information |
|---|---|---|---|---|
| Sequence-based reagent | Mouse *Edem1* | this paper | 3' primer | CGGCAGTAGATGGGGTTGAG |
| Sequence-based reagent | Mouse *Calreticulin* | this paper | 5' primer | CCTGCCATCTATTTCAAAGAGCA |
| Sequence-based reagent | Mouse *Calreticulin* | this paper | 3' primer | GCATCTTGGCTTGTCTGCAA |
| Sequence-based reagent | Mouse *Hyou1* | this paper | 5' primer | TGCGCTTCCAGATCAGTCC |
| Sequence-based reagent | Mouse *Hyou1* | this paper | 3' primer | GGAGTAGTTCAGAACCATGCC |
| Sequence-based reagent | Mouse *Canx* | this paper | 5' primer | ATGGAAGGGAAGTGGTTACTGT |
| Sequence-based reagent | Mouse *Canx* | this paper | 3' primer | GCTTTGTAGGTGACCTTTGGAG |
| Sequence-based reagent | Mouse *GRP94* | this paper | 5' primer | TCGTCAGAGCTGATGATGAAGT |
| Sequence-based reagent | Mouse *GRP94* | this paper | 3' primer | GCGTTTAACCCATCCAACTGAAT |
| Sequence-based reagent | Mouse *Hes1* | this paper | 5' primer | CCAGCCAGTGTCAACACGA |
| Sequence-based reagent | Mouse *Hes1* | this paper | 3' primer | AATGCCGGGAGCTATCTTTCT |
| Sequence-based reagent | Mouse *Deltex* | this paper | 5' primer | ATCAGTTCCGGCAAGACACAG |
| Sequence-based reagent | Mouse *Deltex* | this paper | 3' primer | CGATGAGAGGTCGAGCCAC |
| Sequence-based reagent | Mouse *preTCRa* | this paper | 5' primer | TCACACTGCTGGTAGATGGA |
| Sequence-based reagent | Mouse *preTCRa* | this paper | 3' primer | TAGGCTCAGCCACAGTACCT |
| Sequence-based reagent | Mouse *Notch1* | this paper | 5' primer | ACACTGACCAACAAATGGAGG |
| Sequence-based reagent | Mouse *Notch1* | this paper | 3' primer | GTGCTGAGGCAAGGATTGGA |
| Sequence-based reagent | Mouse *Actin* | this paper | 5' primer | TACCACCATGTACCCAGGCA |

*Continued on next page*

*Appendix 1—key resources table continued*

| Reagent type (species) or resource | Designation | Source or reference | Identifiers | Additional information |
|---|---|---|---|---|
| Sequence-based reagent | Mouse *Actin* | this paper | 3′ primer | CTCAGGAGGAGCAATGATCTTGAT |
| Sequence-based reagent | Vβ5 Forward | this paper | 5′ primer | 5′ CCCAGCAGATTCTCAGTCCAACAG 3′ |
| Sequence-based reagent | Vβ8 Forward | this paper | 3′ primer | 5′ GCATGGGCTGAGGCTGATCCATTA 3′ |
| Sequence-based reagent | Vβ11 Forward | this paper | 5′ primer | 5′ TGCTGGTGTCATCCAAACACCTAG 3′ |
| Sequence-based reagent | Jβ2 Reverse | this paper | 3′ primer | 5′ TGAGAGCTGTCTCCTACTATCGATT 3′ |
| Sequence-based reagent | eF-la Forward | this paper | 5′ primer | 5′CTGCTGAGATGGGAAAGGGCT-3′ |
| Sequence-based reagent | eF-la Reverse | this paper | 3′ primer | 5′ TTCAGGATAATCACCTGAGCA 3′ |
| Sequence-based reagent | Sel1l promoter reporter wildtype | this paper | 5′ primer | TAGCACGCGTGGGAAATGACAAGCGGCATTGTCTTGTAC |
| Sequence-based reagent | Sel1l promoter reporter wildtype | this paper | 3′ primer | ATGCCTCGAGCCTGCTCTCGAAGGTCGAGAGCC |
| Sequence-based reagent | Sel1l promoter reporter mutated left_arm | this paper | 5′ primer | CAGTTCAGGTATAGCTTATGGATCCGCGTTCATATCATGTCCAGTTCAAGGGATCCAAATAATTAAAAAGAAATACTTAGC |
| Sequence-based reagent | Sel1l promoter reporter mutated left_arm | this paper | 3′ primer | GCTAAGTATTTCTTTTTAATTATTTGGATCCCTTGAACTGGACATGATATGAACGCGGATCCATAAGCTATACCTGAACTG |
| Sequence-based reagent | Sel1l promoter reporter mutated right-arm | this paper | 5′ primer | TCTGGGCCAGGGAGGCCGTAAGGGGGGCGAAGAAGGAACC |
| Sequence-based reagent | Sel1l promoter reporter mutated right-arm | this paper | 3′ primer | TCTGGGCCAGGGAGGCCGTAAGGGGGGCGAAGAAGGAACC |
| Sequence-based reagent | Hrd1 promoter reporter wildtype | this paper | 5′ primer | TAGCACGCGTGTGACCCCTGTGTAACGGTTTGATTCC |
| Sequence-based reagent | Hrd1 promoter reporter wildtype | this paper | 3′ primer | ATGCAAGCTTGAAAACAGATATAGGTCTTCC |
| Sequence-based reagent | Hrd1 promoter reporter mutated left arm | this paper | 5′ primer | CCCCGGCCTATGGACTGCGCTGCATACGCTGGCATCCAGCTGCCTTGGCA |
| Sequence-based reagent | Hrd1 promoter reporter mutated left arm | this paper | 3′ primer | TGCCAAGGCAGCTGGATGCCAGCGTATGCAGCGCAGTCCATAGGCCGGGG |

*Continued on next page*

*Appendix 1—key resources table continued*

| Reagent type (species) or resource | Designation | Source or reference | Identifiers | Additional information |
|---|---|---|---|---|
| Sequence-based reagent | Hrd1 promoter reporter mutated middle arm | this paper | 5' primer | CCAGAAATTTTTCCTTTCTTGCATACTTGGTCCGCGTAACTTT |
| Sequence-based reagent | Hrd1 promoter reporter mutated middle arm | this paper | 3' primer | AAAGTTACGCGGACCAAGTATGCAAGAAAGGAAAAATTTCTGG |
| Sequence-based reagent | Hrd1 promoter reporter mutated right arm | this paper | 5' primer | TAGCACGCGTGTGACCCCTGTGTAACGGTTTGATTCC |
| Sequence-based reagent | Hrd1 promoter reporter mutated right arm | this paper | 3' primer | ATGCAAGCTTGAAAACAGATATAGGTCCCTTACGGTTACCTCCCCCCAAC |
| Sequence-based reagent | ChIP-qPCR, Sel1l promoter P1 | this paper | 5' primer | TTCAGTTCAGGTATAGCTTATTTCTCAGCG |
| Sequence-based reagent | ChIP-qPCR, Sel1l promoter P1 | this paper | 3' primer | CGGTTAAGAACTTGCAAGGTTGCTAAG |
| Sequence-based reagent | ChIP-qPCR, Sel1l promoter P2 | this paper | 5' primer | CCTTATGCCCTCAGCCACCTGCGGC |
| Sequence-based reagent | ChIP-qPCR, Sel1l promoter P2 | this paper | 3' primer | GGGAACCCTCATCCAGGACTAC |
| Sequence-based reagent | ChIP-qPCR, Sel1l promoter P3 | this paper | 5' primer | CGCTTAACAAGACAGCTGTTGGG |
| Sequence-based reagent | ChIP-qPCR, Sel1l promoter P3 | this paper | 3' primer | TCTGGGGATTCAAATAACCATCTGGG |
| Sequence-based reagent | ChIP-qPCR, Hrd1 promoter P1 | this paper | 5' primer | GCTAGTTATGAATTGTAAGTAAACGTCTG |
| Sequence-based reagent | ChIP-qPCR, Hrd1 promoter P1 | this paper | 3' primer | CTGATTCTAGACGACTTTAAGGCAG |
| Sequence-based reagent | ChIP-qPCR, Hrd1 promoter P2 | this paper | 5' primer | AACCAATCGGCGGTAGCCACGG |
| Sequence-based reagent | ChIP-qPCR, Hrd1 promoter P2 | this paper | 3' primer | GGATAGCTACGACACGGTAAGAAG |
| Sequence-based reagent | ChIP-qPCR, Hrd1 promoter P3 | this paper | 5' primer | TGCCCAGGTTTCACAGTGCAGC |
| Sequence-based reagent | ChIP-qPCR, Hrd1 promoter P3 | this paper | 3' primer | ACCGAGACGCAGGAGAACACC |
| Sequence-based reagent | ChIP-qPCR, Os9 promoter P1 | this paper | 5' primer | GCTAGAGATGTCCCTTCCGC |

*Continued on next page*

*Appendix 1—key resources table continued*

| Reagent type (species) or resource | Designation | Source or reference | Identifiers | Additional information |
|---|---|---|---|---|
| Sequence-based reagent | ChIP-qPCR, Os9 promoter P1 | this paper | 3' primer | CAGCCAATGAAAGCTTGGGG |
| Sequence-based reagent | ChIP-qPCR, Os9 promoter P2 | this paper | 5' primer | GGAGGATAGCCGTGCTTTGA |
| Sequence-based reagent | ChIP-qPCR, Os9 promoter P2 | this paper | 3' primer | ATCATAGCTAAGGAGTGAGAATGAG |
| Sequence-based reagent | ChIP-qPCR, Edem1 promoter P1 | this paper | 5' primer | CTACTCCATACCTGGACGGG |
| Sequence-based reagent | ChIP-qPCR, Edem1 promoter P1 | this paper | 3' primer | GCCCTAGCCCGGGTAAATG |
| Sequence-based reagent | ChIP-qPCR, Edem1 promoter P2 | this paper | 5' primer | CCCTGGTGAGTTGCTGATGT |
| Sequence-based reagent | ChIP-qPCR, Edem1 promoter P2 | this paper | 3' primer | TGCTGTGAGTGTGTATGCGT |
| Sequence-based reagent | Mouse Xbp1 splicing | PMID:29480818 | 5' primer | ACACGCTTGGGAATGGACAC |
| Sequence-based reagent | Mouse Xbp1 splicing | PMID:29480818 | 3' primer | CCATGGGAAGATGTTCTGGG |
| Sequence-based reagent | Mouse *Actin* | PMID:29480818 | 5' primer | TACCACCATGTACCCAGGCA |
| Sequence-based reagent | Mouse *Actin* | PMID:29480818 | 3' primer | CTCAGGAGGAGCAATGATCTTGAT |
| peptide, recombinant protein | Recombinant Murine Flt3-Ligand, 2 ug | 250–31L | peprotech | |
| peptide, recombinant protein | Recombinant Murine IL7, 2 ug | 217–17 | peprotech | |
| peptide, recombinant protein | Recombinant Mouse DLL4 | 776702 | Biolegend | |
| commercial assay or kit | High-Capacity cDNA Reverse Transcription Kit | 4368813 | Thermo Fisher | |
| commercial assay or kit | Power SYBR Green PCR Master Mix | A25778 | Thermo Fisher | |
| commercial assay or kit | In Situ Cell Death Detection Kit, Fluorescein | 11684795910 | Roche | |
| commercial assay or kit | FITC BrdU Flow Kit | 559619 | BD Bioscience | |

*Continued on next page*

*Appendix 1—key resources table continued*

| Reagent type (species) or resource | Designation | Source or reference | Identifiers | Additional information |
|---|---|---|---|---|
| commercial assay or kit | eBioscience Foxp3 / Transcription Factor Staining Buffer Set | 00-5523-00 | Thermo Fisher | |
| commercial assay or kit | ER-Tracker Green (BODIPY FL Glibenclamide), for live-cell imaging | E34251 | Thermo Fisher | |
| commercial assay or kit | ProteoStat (R) Aggresome detect Kit | ENZ-51035-K100 | ENZO | |
| commercial assay or kit | Click-iT Plus OPP Alexa Fluor 647 Protein Synthesis Assay Kit | C10458 | Thermo Fisher | |
| commercial assay or kit | Genomic DNA Mini Kit | K182002 | Thermo Fisher | |
| commercial assay or kit | Truseq Stranded mRNA Kit | # 20020594 | Illumina | |
| Chemical compound, drug | DAPT | HY-13027 | MedChemExpress | |
| Chemical compound, drug | ISRIB (trans-isomer) | HY-12495 | MedChemExpress | |
| Chemical compound, drug | Tunicamycin | 76102–666 | VWR | |
| Chemical compound, drug | Tetraphenylethene maleimide (TMI) | PMID:31914399 | Custom Synthesized | |
| Software, algorithm | STAR | | | Version 2.5.2b |
| Software, algorithm | DESeq2 | | R 3.6.1 | Version 1.26.0 |
| Software, algorithm | fgsea | | R 3.6.1 | Version 1.11.2 |
| Software, algorithm | pheatmap | | R 3.6.1 | Version 1.0.12 |
| Software, algorithm | Enrichr | | | https://maayanlab.cloud/Enrichr/ |
| Software, algorithm | Graphpad Prism 8.4 | Graphpad (graphpad.com) | RRID:SCR_002798 | |
| Software, algorithm | Fiji | http://imagej.net/Fiji | RRID:SCR_003070 | |
| Software, algorithm | FlowJo 10 | Tree Star | RRID:SCR_008520 | |
| Software, algorithm | Biorender | https://biorender.com/ | | Biorender was utilized to make the schematic diagrams used in this study. |

*Continued on next page*

*Appendix 1—key resources table continued*

| Reagent type (species) or resource | Designation | Source or reference | Identifiers | Additional information |
|---|---|---|---|---|
| Other | LD Columns | 130-042-901 | Miltenyi Biotec | |
| Other | CD4 (L3T4) MicroBeads, mouse 1 x 2 mL | 130-117-043 | Miltenyi Biotec | |
| Other | CD8a (Ly-2) MicroBeads, mouse | 130-117-044 | Miltenyi Biotec | |
| Other | DAPI (4',6-Diamidino-2-Phenylindole, Dihydrochloride) | D1306 | Thermo Fisher | |
| Other | Precision Count Beads | 424902 | Biolegend | |
| Other | RBC Lysis Buffer (10X) | 420301 | Biolegend | |
| Other | Liquid DAB+ | K3468 | Dako | |

