## [Decision Letter]

**Acceptance summary:**

The findings by Liu and colleagues will be of interest to those working on T cell development, thymus biology and metabolic regulation of lymphocyte differentiation. The authors show that thymocytes undergoing pre-T cell receptor signaling, which leads to a highly proliferative state, require the endoplasmic reticulum-associated degradation (ERAD) complex to maintain high fidelity in their protein translational machinery, which enables thymocytes to effectively undergo further differentiation. Key components of the ERAD complex are shown to be under Notch regulation, further extending the absolute and critical role for Notch signaling in the differentiation of thymocytes.

**Decision letter after peer review:**

Thank you for submitting your article "Notch-Induced Endoplasmic Reticulum-Associated Degradation Governs Thymocyte b-Selection" for consideration by *eLife*. Your article has been reviewed by 3 peer reviewers, including JC Zúñiga-Pflücker as the Reviewing Editor and Reviewer #1, and the evaluation has been overseen Satyajit Rath as the Senior Editor. The following individuals involved in review of your submission have agreed to reveal their identity: Michele Anderson (Reviewer #2); David Wiest (Reviewer #3).

Essential revisions:

1) All three reviewers concurred with the importance and novelty of the work. No new experiments are required, and only clarifications or additional discussion points are required. Please see below for the specific points that need to be addressed.

*Reviewer #1:*

The work by Liu et al. addresses the translational activity associated with the β-selection step in T cell development, during which CD4- CD8- double negative (DN) thymocytes that have properly rearranged a T cell receptor (TCR) β chain undergo several quick rounds of cell division. The requirement for Notch and TCR-β, along with a pre-T-α chain to form the pre-TCR, have been understood for some time. However, whether Notch also regulated aspects of the translational activity in β-selection remained unknown.

Here the authors demonstrate that thymocyte differentiation from the DN to the CD4^+^ CD8^+^ double positive (DP) stage is tightly regulated by endoplasmic reticulum (ER)-associated degradation (ERAD) complex, which ensures proteome homeostasis, and key components of the ERAD are themselves under transcriptional regulation by Notch. These conclusions are solidly supported by several genetic approaches that take advantage of conditional knock-out mice for Sel1l, a key component of the ERAD complex.

The authors show that Sel1l deficient thymocytes fail to undergo efficient β-selection events and have increased ER stress and apoptosis via an unfolded protein response (UPR) activation of the PERK pathway. An elegant mixed bone marrow chimera approach is used to demonstrate the intrinsic requirement for Sel1l in allowing for effective β-selection.

The authors also reveal that the XBP1-dependent pathway play a redundant role during β-selection. Overall, the conclusions are strongly supported by the experimental approaches.

To increase the breath and potential impact of their findings the authors are encouraged to discuss whether a similar requirement for Sel1l would be present in T-ALLs that show a similar need for Notch signaling to maintain their proteostasis.

The work presented by the authors convincingly illustrates the role played by the ERAD complex in enabling β-selection induced proliferation and associated protein synthesis. The demonstration that Notch signaling directly regulates several of the key components of the ERAD complex, such as Sel1l1, nicely adds to our understanding of the critical role play by Notch at this stage of T cell development.

1) Given that Notch is also critically important for the survival and metabolism of many T-ALLs, it would be good for the authors to add this consideration in their discussion. This would further the impact of their work. Plus, if possible, they may be able to show that interfering with the Sel1l-HRD1 pathway can lead to T-ALL cell apoptosis.

2) It is not clear how the Dll4 signaling experiments were performed, was the Dll4 protein pre-bound to the plates prior to adding the EL4 cells? Otherwise, it is quite surprising to see a soluble Notch ligand being able to induce a productive Notch signal.

3) The dual loss of Sel1l and Xbp1 shows a very dramatic loss in thymus cellularity, perhaps this finding should be mentioned in the abstract.

4) In Figure 2 – supplement 2J, is there are reason that donor derived B cells were not shown?

5) The results shown in Figure 1 – supplement 1D-L, represent an enormous amount of work for the authors, even if the results from three additional conditionally-deleted gene mouse models are negative, they should be commended for their thoroughness.

*Reviewer #2:*

This manuscript describes a new type of quality control during the TCRbeta(b)-selection checkpoint of early T cell development. It is known that pre-TCR and Notch signaling at the DN3 to DN4 stages of thymocyte development results in a proliferative burst. High rates of protein translation during this rapid expansion can overwhelm the machinery required for proper protein folding, leading to endoplasmic reticulum (ER) stress. ER stress induces several pathways including the endoplasmic reticulum (ER)-associated degradation (ERAD) pathway and the unfolded protein response (UPR) pathways. While these pathways have been well explored in plasma cell development, their roles in T cell development have been unclear. The importance of this question for understanding thymocyte biology is a major strength of the paper.

In this study, the authors conditionally deleted Sel1l, a central mediator of ERAD, in early T cell precursors using a CD2-Cre transgene. They found that thymocytes beyond the b-selection checkpoint were severely decreased, indicating a requirement for the ERAD pathway. Using a diverse set of complementary assays, they showed that the absence of Sel1l led to increased apoptosis of DN3 cells due to an increase in unfolded and misfolded proteins. This conclusion was robustly supported by separate assays measuring protein synthesis, ER stress, protein aggregation, proliferation, and apoptosis.

The defect in T cell development at the DN3 to DN4 transition was even more clearly manifested in a competitive bone marrow chimera study, emphasizing a cell intrinsic role for Sel1l in maintaining DN3 fitness.

By contrast, loss of Sel1l in the CD2-Cre mice did not affect gd T cell development, although the authors did not distinguish between different gd T cell subsets in this study. They also found that Sel1l was dispensable for T cell development after b-selection using a CD4-Cre transgene, which was consistent with the fact that positive selection of DP thymocytes does not result in a proliferative burst. The specific importance of the ERAD pathway was reinforced by studies of a number of mouse strains carrying mutations in the UPR responses, which did not perturb T cell development. The comprehensive nature of these analyses is a major strength of the paper.

Another important finding was that multiple members of the ERAD pathway were under direct regulation of Notch signaling in T lineage cells, providing a previously unappreciated role for Notch1 in regulating survival. Again, multiple complementary assays were used to reach this conclusion, including chromatin immunoprecipitation, promoter assays, and the use of OP9-DL4 cells to activate Notch signaling, which is a much more physiological method for studying Notch biology than introducing a constitutively activated form of Notch.

Interestingly, an RNA-seq analysis of Sell1l-deficient DN3 cells showed that genes belonging to UPR pathways were upregulated, presumably to help compensate for the absence of ERAD. Double mutants of Sel1l and Xbp1 exacerbated the defect, whereas deletion of PERK in Sel1l-deficient mice restored thymocyte numbers and survival. Thus, the authors concluded that Notch-mediated upregulation of the ERAD machinery alleviates the proteotoxic stress that occurs during the post-b selection expansion, and that PERK is required for the ER-stress cell death that occurs in the absence of ERAD.

This is a well-written, elegant, and comprehensive study that sheds new light on physiological parameters of T cell development. Moreover, the connection between Notch1 and ERAD highlights its central role in many aspects of survival at the b-selection checkpoint.

This is a comprehensive analysis of the pathways that regulate proteotoxic stress during T cell development, with well supported conclusions and beautifully presented data.

*Reviewer #3:*

The manuscript by Liu et al., explores the mechanisms controlling proteostasis during development of T cell progenitors in the thymus. Their principal finding is that the rapid transition from quiescence to active proliferation that occurs during traversal of the β-selection checkpoint causes the accumulation of misfolded proteins that are managed via dislocation from the endoplasmic reticulum (ER) into the cytol and degradation through the process of ER-associated degradation (ERAD). Moreover, the trophic signals that promote this transition and the capacity of these cells to manage their proteotoxic insult are both linked to Notch signaling and its capacity to induce genes associated with ERAD, specifically Sel1l. Sel1l appears to be a direct Notch target and its absence blocks traversal of the β-selection checkpoint via induction of proteotoxicity induced apoptosis. Apoptosis and the developmental arrest can be alleviated by co-elimination of one of the major ER-stress signaling effectors, PERK, but not another ER-stress effector XBP-1. Based on these findings, the authors conclude that Notch regulation of ERAD, but not ER-stress (or the unfolded protein response) play a critical role in supporting traversal of the β-selection checkpoint.

This study is well designed and executed, the experimental findings support the authors conclusions, and work performed provides new insight into a new role for Notch signaling in supporting the development of α-β lineage T cells.

---

## [Author Response]

Reviewer #1:[…] To increase the breath and potential impact of their findings the authors are encouraged to discuss whether a similar requirement for Sel1l would be present in T-ALLs that show a similar need for Notch signaling to maintain their proteostasis.The work presented by the authors convincingly illustrates the role played by the ERAD complex in enabling β-selection induced proliferation and associated protein synthesis. The demonstration that Notch signaling directly regulates several of the key components of the ERAD complex, such as Sel1l1, nicely adds to our understanding of the critical role play by Notch at this stage of T cell development.1) Given that Notch is also critically important for the survival and metabolism of many T-ALLs, it would be good for the authors to add this consideration in their discussion. This would further the impact of their work. Plus, if possible, they may be able to show that interfering with the Sel1l-HRD1 pathway can lead to T-ALL cell apoptosis.

We thank the reviewer for raising this important point. We have now discussed the potential implication of ERAD in Notch-mediated T-ALLs in Discussion section as below:

“Activating NOTCH1 mutations are found in more than 50% of human T cell acute lymphoblastic leukemia (T-ALL) patients wherein Notch plays pivotal roles in regulating the survival and metabolism of T-ALL cells. It will be of interest in future studies to investigate whether Notch-regulated ERAD also promotes ER proteostasis and sustains T-ALL survival by clearing misfolded proteins and restricting ER stress-induced cell death".

2) It is not clear how the Dll4 signaling experiments were performed, was the Dll4 protein pre-bound to the plates prior to adding the EL4 cells? Otherwise, it is quite surprising to see a soluble Notch ligand being able to induce a productive Notch signal.

For Figure 4C, DLL4 was pre-bound to the plates prior to adding the EL4 cells. For Figure 4B, F-I, K, and L, soluble DLL4 (5mg/ml) was used. We agree with the reviewer that pre-bound Notch ligands tether the Notch extracellular domain away from the NICD. However, other studies also show that 0 Notch ligands can induce a productive Notch signal as well (PMID: 22869002; PMID: 8575327; PMID: 12370358)^1–3^. In PMID:22869002^1^, both anchored and soluble DLL4 induced HES^-1^ and Notch cleavage of the intracellular domain significantly in macrophages. We used exactly the same DLL4 and our data show that it effectively induced classical Notch target genes *Hes1*, *Deltex1* and *Ptcra* (Figure 4—figure supplement 1A). We have clarified the technical details in Materials and methods.

3) The dual loss of Sel1l and Xbp1 shows a very dramatic loss in thymus cellularity, perhaps this finding should be mentioned in the abstract.

We appreciate the reviewer’s kind suggestion. We have now mentioned this finding in the Abstract as below:

“In contrast, IRE1a/XBP1 pathway was induced as a compensatory adaptation to alleviate *Sel1l*-deficiency induced ER stress. Dual loss of *Sel1l* and *Xbp1* markedly exacerbated the thymic defect”.

4) In Figure 2 – supplement 2J, is there are reason that donor derived B cells were not shown?

We thank the reviewer for the questions and have now included donor derived B cells data in Figure 2-supplement 2J. *hCD2-iCre* efficiently induces gene deletion in B cells (PMID: 12548562)^4^. Consistent with previous publications showing that SEL1L/HRD1 ERAD is required for B cell development through regulating pre-BCR (PMID: 27568564 and 29907570)^5,6^, our *hCD2-iCre* mediated *Sel1l* deletion also caused significant B cell reconstitution defects in the recipient mice. We have included the data and description in the manuscript.

5) The results shown in Figure 1 – supplement 1D-L, represent an enormous amount of work for the authors, even if the results from three additional conditionally-deleted gene mouse models are negative, they should be commended for their thoroughness.

We appreciate the reviewer’s kind comments on our data from multiple UPR conditional knockout mouse models.

References:

1. Camelo, S. et al. Δ-like 4 inhibits choroidal neovascularization despite opposing effects on vascular endothelium and macrophages. Angiogenesis 15, 609–622 (2012).

2. Fitzgerald, K. and Greenwald, I. Interchangeability of *Caenorhabditis elegans* DSL proteins and intrinsic signalling activity of their extracellular domains in vivo. Dev Camb Engl 121, 4275–82 (1995).

3. Weijzen, S. et al. The Notch Ligand Jagged-1 Is Able to Induce Maturation of Monocyte-Derived Human Dendritic Cells. J Immunol 169, 4273–4278 (2002).

4. Boer, J. de et al. Transgenic mice with hematopoietic and lymphoid specific expression of Cre. Eur J Immunol 33, 314–325 (2003).

5. Ji, Y. et al. The Sel1L-Hrd1 Endoplasmic Reticulum-Associated Degradation Complex Manages a Key Checkpoint in B Cell Development. Cell Reports 16, 2630–2640 (2016).

6. Yang, Y. et al. The endoplasmic reticulum–resident E3 ubiquitin ligase Hrd1 controls a critical checkpoint in B cell development in mice. J Biol Chem 293, 12934–12944 (2018).